# Skill-Driven Neurosymbolic State Abstractions

**Alper Ahmetoglu**[*]
Brown University

**Steven James**
University of the Witwatersrand

**Cameron Allen**
UC Berkeley

**Sam Lobel**
Brown University

**David Abel**
University of Edinburgh

**George Konidaris**
Brown University

## Abstract

We consider how to construct state abstractions compatible with a given set of abstract actions, to obtain a well-formed abstract Markov decision process (MDP). We show that the Bellman equation suggests that abstract states should represent distributions over states in the ground MDP; we characterize the conditions under which the resulting process is Markov and approximately model-preserving, derive an algorithm for constructing the abstract MDP, and apply it to visual chain and maze tasks. We generalize these results to the factored actions case, characterize the conditions that lead to factored abstract states, and apply the resulting algorithm to a visual grid and Montezuma's Revenge. These results provide a principled, powerful framework for learning neurosymbolic abstract Markov decision processes.

## 1 Introduction

Reinforcement learning (RL) [62] has achieved remarkable success solving tasks with complex sensorimotor spaces [48, 43]. However, learning and planning at the pixel and motor level, while necessary, must eventually be insufficient: the complexity of real-world tasks such as cooking a meal, inter-city travel, and fixing a car is such that they can only be feasible if decision-making occurs at an abstract level and the low-level details are abstracted away. This is especially true when a single agent must perform *all* these tasks, necessitating a sensorimotor space complex enough to cover all of them. Linking abstract reasoning to low-level action and perception, to enable fast decision-making grounded in real sensorimotor interaction, is therefore critical for general intelligence [23, 38].

Hierarchical RL (HRL) [14] addresses this problem via high-level abstract *actions*, but does not create correspondingly abstract *states*, leaving the agent in its original low-level state space. It would be far preferable to combine *both* types of abstractions to construct an entirely abstract decision process [38], which requires mutually compatible state and action abstractions: those that result in a new, more compact decision process that is Markov and supports near-optimal policies over the abstract actions in the original MDP. Following earlier work in robotics [40], we adopt an *action-first* strategy, constructing state abstractions compatible with a pre-existing set of abstract actions.

We first focus on building an abstract MDP from a ground MDP augmented with a given set of abstract actions, where the Bellman equation suggests that abstract states should represent distributions over ground states. We use this insight to build an abstract decision process, and identify the conditions under which it is Markov and approximately model-preserving [44, 1]. We develop algorithms for constructing the abstraction from data and for planning with it, and apply them to visual chainwalk and maze tasks. We then generalize these results to factored actions, which modify only some state variables in the ground MDP. We characterize the conditions under which this generates factored

---

[*]Correspondence to Alper Ahmetoglu `aahmetog@cs.brown.edu` and George Konidaris `gdk@brown.edu`. Extended results are available at `neurosymbolic-mdps.github.io`.

39th Conference on Neural Information Processing Systems (NeurIPS 2025).

abstract states, provide an algorithm that constructs the corresponding abstraction, and apply it to a visual gridworld and Montezuma's Revenge, a long-horizon Atari task [15]. These results provide a powerful and principled framework for learning neurosymbolic abstract decision processes.

## 2 Background

RL problems are typically formalized as *Markov decision processes* (MDPs) [62], given by a tuple $M = (S, A, R, T, \gamma)$, where $S$ is a set of states; $A$ is a set of actions, with $A(s) \subseteq A$ denoting the actions available at state $s$; $R(s, a, s')$ is the reward received when executing action $a \in A(s)$ in state $s$ and transitioning to state $s'$; $T(s'|s, a)$ is the probability of entering state $s'$ upon execution action $a$ in state $s$; and $\gamma \in [0, 1]$ is a discount factor. A decision process is *Markov* when: 1) State $s$ is sufficient for determining which actions are executable; i.e., $A(s)$ is only a function of $s$. 2) State $s$ and action $a$ are sufficient for determining the probability of transitioning to state $s'$; i.e., the distribution $T(s'|s, a)$ is conditionally independent of all other variables. 3) State $s$, action $a$, and successive state $s'$ determine reward, so $R(s, a, s')$ depends on no other variables. The state space of a task must therefore support Markov reward ($R$), transition ($T$), and available action ($A$) functions.

We focus on the multi-task RL setting where the agent must solve several MDPs drawn from a task distribution. We model this as a fixed *ground MDP* to which absorbing goal states (with corresponding rewards) are superimposed to create new tasks. We define the ground MDP $M_0$ as: $M_0 = (S_0, A_0, R_0, T_0, \gamma)$, with state space $S_0$, action space $A_0$, transition function $T_0$, discount factor $\gamma$, and background reward function $R_0$. Any individual task $\tau \sim P(\tau)$ is given by a *goal state set* $G_\tau$ and an additional goal reward function $R_\tau$ that is non-zero only for transitions entering $G_\tau$.

**Action Abstraction**    *Temporal* or *action abstractions* are constructed over the primitive action set $A_0$ [14]. Although other formalizations exist [22, 49] (see Klissarov et al. [36] for a comprehensive overview), we adopt the *options framework* [64], where the agent has a collection of options $O$, where each $o \in O$ is a tuple $(I_o, \beta_o, \pi_o)$. The *initiation set*, $I_o \subseteq S$, is the set of states from which $o$ may be invoked. It is often defined as a classifier $I_o : S \rightarrow \{0, 1\}$. To refer to the availability of all options at a given state, we define $I_O : S \rightarrow \{0, 1\}^{|O|}$ to be a vector with entry $i$ set to $I_{o_i}(s)$. Instances of the *initiation vector* are written as $I$. Similarly, the *termination condition*, $\beta_o : S \rightarrow [0, 1]$, is the probability of option termination when entering $s$. $\beta_O : S \rightarrow [0, 1]^{|O|}$ is the *termination vector*: a vector of every option's termination probability at state $s$. Finally, *option policy*, $\pi_o : S \rightarrow A$, controls option execution, which begins from a state in $I_o$, follows $\pi_o$, and terminates with probability $\beta_o(s_t)$ at each state $s_t$.

Replacing the primitive actions of an MDP with options results in a *semi-Markov decision process* (or SMDP) $M = (S, O, R, T, \gamma)$, where now $O$ is a set of options, with $O(s) = \{o \in O | s \in I_o\}$ denoting the options available in state $s$; $R(s, o, s')$ returns the expected summed discounted reward received when executing option $o \in O(s)$ at state $s \in S$ and exiting in $s' \in S$; $T(s', t|s, o)$ describes probability of arriving in $s' \in S$, $t$ time steps after executing $o \in O(s)$ from $s \in S$. The *expected-length* option model [3] represents these quantities separately, using transition function $T(s'|s, o)$ and discount model $\gamma(s, o) = \gamma^\tau$, where $\tau$ is the expected transition length. An *option model* for $o$ consists of initiation set $I_o$, reward model $R$, transition model $T$, and discount model $\gamma$. This representation is sufficient for model-based planning [64, 57, 63] and can be substantially faster than low-level planning when options shorten the effective solution depth [60, 50]. An agent not given a collection of options must discover them itself. *Skill discovery algorithms*—an active area of research [27]—typically identify $\beta_o$ and then learn the remaining components to successfully reach it. Our framework is intentionally agnostic to the origin of the agent's available options.

**State Abstraction**    State abstraction [44] attempts to reduce the state space size by discarding irrelevant information or compressing the state space. Previous work has largely defined a state abstraction using an *abstraction function*: $\phi : S \rightarrow \overline{S}$ mapping each state in the original state space $S$ to a state in a new abstract state space $\overline{S}$. This approach is often called state aggregation [44]. Although there has been some work on option-specific state abstractions for efficient option policy learning [32, 39, 19], the two abstraction types are typically considered separately.

# 3 Skill-Driven Abstract MDPs

Given a ground MDP $M = (S, A, R, T, \gamma)$, our goal is to construct an abstract MDP $\overline{M} = (\overline{S}, \overline{A}, \overline{R}, \overline{T}, \gamma)$. The core assumption behind almost all HRL is that $\overline{A}$ is a set of options defined over $M$ but typically $\overline{S} \equiv S$, missing the opportunity to build a completely abstract MDP.

While we could attempt state and action abstraction independently and hope that the resulting MDP is well-formed, that seems unlikely to succeed, so one type of abstraction must be constructed to support the other.[2] Although the reverse is certainly possible (see Section 5), consider the properties that make a state Markov: it must support determining which actions can be executed, the distribution over successor states given an action, and the reward. *An observation space qualifies as a state space solely by virtue of supporting these functions, which all model the operations of actions,* and an abstract state space that satisfies these properties for a given option set leads to a well-formed MDP. Our approach therefore constructs the state abstraction to support the action abstraction.

## 3.1 The Semantics of Abstract States

One important question that must be answered when constructing an abstract state space is one of *semantics*: what relationship should hold between an abstract state and the ground MDP? Previous work categorizes state abstraction types according to which properties they ensure hold in the ground MDP [44]. Instead, we adopt a *constructive* approach [40] by defining a target computation and then designing an abstraction that supports it by construction. To support as wide a range of learning algorithms as possible, we target the Bellman equation [16]—the basis of virtually all value-function-based RL algorithms—which for evaluating policy $\pi$ (over options) at state $s$ gives:

$$\begin{aligned} V(s) &= \sum_{\overline{a}} \pi(\overline{a}|s)\mathbb{E}\left[r + \gamma^\tau V(s')\right] = \sum_{\overline{a}} \pi(\overline{a}|s)\left(\mathbb{E}\left[R(s,\overline{a},s')\right] + \mathbb{E}\left[\gamma^\tau V(s')\right]\right) \\ &\approx \sum_{\overline{a}} \pi(\overline{a}|s)\left(\mathbb{E}\left[R(s,\overline{a},s')\right] + \gamma^{\mathbb{E}[\tau]}\mathbb{E}\left[V(s')\right]\right), \end{aligned} \tag{1}$$

where expectations are over $s'$ and $\tau$ given $(s, \overline{a})$, and the approximation is the expected-length model [3]. Notice that the last equation is primarily composed of *expectations over distributions of states*.

If we refer to a distribution over states using a bar (e.g. $\overline{s}$), the corresponding expected value and rewards are: $V(\overline{s}) = \mathbb{E}[V(s)]$, $R(s, \overline{a}, \overline{s}') = \mathbb{E}[R(s, \overline{a}, s')]$, and $R(\overline{s}, \overline{a}, \overline{s}') = \mathbb{E}[\mathbb{E}[R(s, \overline{a}, s')]]$. Using these definitions, Equation 1 is:

$$V(s) \approx \sum_{\overline{a}} \pi(\overline{a}|s)[R(s, \overline{a}, \overline{s}') + \gamma^{\overline{\tau}}V(\overline{s}')],$$

where $\overline{\tau} = \mathbb{E}[\tau]$ and $\overline{s}' = T(s'|s, \overline{a})$. Later, we will pursue desirable properties by decomposing (or *refining*) $T(s'|s, \overline{a})$ into a mixture distribution, i.e., a weighted sum of component distributions. Figure 1 visualizes such a refinement.

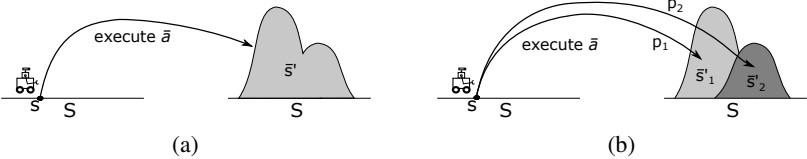

Figure 1: The outcome of executing option $\overline{a}$ from $s$ is described by distribution over states $\overline{s}'$ (a). $\overline{s}'$ can be refined into a mixture of component distributions $\overline{s}'_1$ and $\overline{s}'_2$ with corresponding probabilities $p_1$ and $p_2$, such that $\overline{s}' = p_1\overline{s}'_1 + p_2\overline{s}'_2$ and $p_1 + p_2 = 1$ (b).

Refining $\overline{s}'$ into component distributions $\overline{s}'_i$, i.e., $T(s'|s, \overline{a}) = \sum_i p_i\overline{s}'_i$ where $\sum_i p_i = 1$, obtains:

$$V(s) \approx \sum_{\overline{a}} \pi(\overline{a}|s) \sum_i p_i \left[R(s, \overline{a}, \overline{s}'_i) + \gamma^{\overline{\tau}_i}V(\overline{s}'_i)\right]. \tag{2}$$

Equation 2 is reminiscent of the Bellman equation but contains states of two types: individual states ($s$) on the left side and distributions over states ($\overline{s}_i$) on the right. We next extend the left side of Equation 2 to refer to a distribution of states, computing $V(\overline{s})$ for distribution $\overline{s}$, which is required to

---

[2]That does not mean that one process completely precedes the other in *time*; rather, one *logically* precedes the other—one type of abstraction is designed to match some property of the other.

recursively compute the right side of Equation 2. This is a strict generalization since any state can be viewed as a Dirac delta distribution. Substituting the definition of $V(\bar{s})$:

$$V(\bar{s}) \approx \mathbb{E}_{s \sim \bar{s}} \left[ \sum_{\bar{a}} \pi(\bar{a}|s) \sum_i p_i [R(s, \bar{a}, \bar{s}'_i) + \gamma^{\tau_i} V(\bar{s}'_i)] \right].$$

This resemblance to the original Bellman equation suggests that we should choose abstract states $\bar{s}$ that correspond to distributions over ground states. If we choose to do so, then an abstract policy $\bar{\pi}$ is a function of $\bar{s}$ rather than $s$, so we can shift it out of the expectation to obtain:

$$V(\bar{s}) \approx \sum_{\bar{a}} \bar{\pi}(\bar{a}|\bar{s}) \sum_i p_i \left[ R(\bar{s}, \bar{a}, \bar{s}'_i) + \gamma^{\tau_i} V(\bar{s}'_i) \right]. \tag{3}$$

Equation 3 is the original Bellman equation applied to abstract states and actions, and is the sole equation necessary for computing Bellman updates. In other words, *the Bellman equation for a given set of options can therefore be naturally rewritten as a Bellman equation between abstract states, provided each abstract state refers to a distribution over ground states.* It remains to determine how to refine these state distributions so that the Markov property holds (that is, the model is an MDP), and $\bar{\pi}^*$ operates with minimal value loss over $M$.

### 3.2 Satisfying the Markov Property

The three Markov properties we seek are: 1) the abstract transition probability and discount models can be written as $T(\bar{s}'|\bar{s}, \bar{a})$ and $\gamma(\bar{s}, \bar{a})$; 2) the abstract reward function can be written as $R(\bar{s}, \bar{a}, \bar{s}')$; and 3) the options executable at an abstract state can be written as $A(\bar{s})$. The first two hold by construction but the third does not, because the distribution over states resulting from an option execution may contain states from which different options are executable. Therefore, preserving the Markov property in the abstract MDP requires that state distributions be refined (at least) into component distributions from which a consistent set of options can be executed. Figure 2 gives an example distribution and initiation set combination where this occurs and the refinement that resolves it. If the initiation sets are known exactly the component distributions do not overlap;

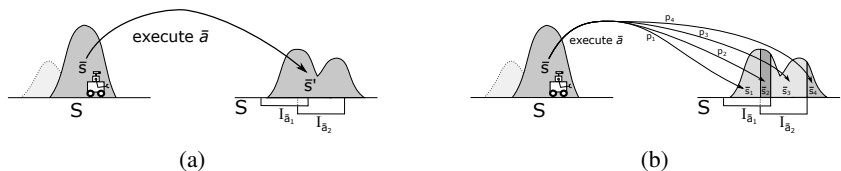

(a)                          (b)

Figure 2: Option execution where the distribution over next states cannot specify which options are available next. (a) Executing $\bar{a}$ from distribution $\bar{s}$ leads to state distribution $\bar{s}'$. Some states in $\bar{s}'$ are in initiation set $I_{\bar{a}_1}$, some in $I_{\bar{a}_2}$, some in both and some neither. Therefore $\bar{s}'$ is not a Markov state as it does not specify available actions. (b) $\bar{s}'$ can be refined into distributions $\bar{s}_1 - \bar{s}_4$ (occurring with probability $p_1 - p_4$), each over states with the same available options—only $\bar{a}_1$ can be executed in $\bar{s}_1$, both $\bar{a}_1$ and $\bar{a}_2$ can be executed in $\bar{s}_2$, only $\bar{a}_2$ in $\bar{s}_3$, and neither in $\bar{s}_4$.

when initiation sets are uncertain (i.e., we only have a probabilistic estimate of when an option is executable), they may have overlapping support. In either case, each component distribution now represents *the distribution over states conditioned on a specific initiation set vector being observed*, so the semantics of abstract states are:

**Definition 1.** An **abstract state** $\bar{s}$ is a tuple $(p, I)$, where $p(s \mid I)$ is a distribution over ground states conditioned on observed initiation vector $I$. We define a *grounding function* for each component: $\mathcal{G}_p(\bar{s}_i) = p_i$ and $\mathcal{G}_I(\bar{s}_i) = I_i$.

We can now form an abstract MDP $\overline{M} = (\overline{S}, \overline{A}, \overline{R}, \overline{T}, \bar{\gamma})$ constructed over ground MDP $M$ a follows. The actions $\overline{A}$ are a set of options over $M$, and the state set $\overline{S}$ is all abstract states over $S$: $\overline{S} = \{\bar{s} \mid \exists p, I \ \mathcal{G}_p(\bar{s}) = p, \mathcal{G}_I(\bar{s}) = I\}$. The transition function is:

$$\overline{T}(\bar{s}'|\bar{s}, \bar{a}) = \begin{cases} P(I' \mid \bar{s}, \bar{a}) & \text{if } \mathcal{G}_p(\bar{s}') = P(s' \mid \bar{s}, \bar{a}, I') \text{ and } \mathcal{G}_I(\bar{s}') = I'. \\ 0 & \text{otherwise,} \end{cases}$$

where $P(I' \mid \bar{s}, \bar{a})$ is the expected probability of observing initiation vector $I'$ after executing $\bar{a}$ from $s \sim \mathcal{G}_p(\bar{s})$, and $P(s' \mid \bar{s}, \bar{a}, I')$ is the expected distribution over $s'$ after executing $\bar{a}$ from

$s \sim \mathcal{G}_p(\bar{s})$ and observing $I'$. The transition function expresses the probability that executing an option from abstract state $\bar{s}$ will lead to various abstract states $\bar{s}'$, each representing a distribution over states conditioned on a specific initiation vector post-execution. Similarly the discount is $\overline{\gamma}(\bar{s}, \overline{a}) = \mathbb{E}_{s \sim \mathcal{G}_p(\bar{s})} \left[ \gamma(s, \overline{a}) \right]$ and the reward function is: $\overline{R}(\bar{s}, \overline{a}, \bar{s}') = \mathbb{E}_{s \sim \mathcal{G}_p(\bar{s})} \left[ \mathbb{E}_{s' \sim \mathcal{G}_p(\bar{s}')} \left[ R(s, \overline{a}, s') \right] \right]$.

## 3.3 Building a Model-Preserving Abstract MDP

The above MDP is Markov by construction and supports computing the exact expected-length value of any start state distribution—because Equations 1 and 3 are equal—with two caveats: there is no obvious way to build the uncountably infinite state space, and only abstract policies $\pi(\overline{a}|\bar{s})$ are supported, when further refinement might do better. *Approximately model-preserving abstractions* [1] support computing approximately optimal policies in the multi-task setting. An abstract MDP is approximately model preserving if:

$$\int_{s'} |T(s'|s, \overline{a}) - T(s'|\bar{s}, \overline{a})| \, ds' \leq \epsilon_T \text{ and } |R(s, \overline{a}, \bar{s}') - R(\bar{s}, \overline{a}, \bar{s}')| \qquad \leq \epsilon_R,$$

for any option $\overline{a}$, pair $(\bar{s}, \bar{s}')$, ground state $s \sim \bar{s}$, and $\epsilon_R, \epsilon_T \geq 0$. Since an approximately model-preserving abstraction is also approximately value- and policy-preserving [1, 45], the optimal policy computed for an abstract MDP that obeys these conditions is also a good approximation to the optimal policy over the options in the ground MDP; indeed:

**Theorem 3.1.** $\forall \bar{s}, \overline{a}, \pi : S \to \overline{A}$, and $s \sim \bar{s}$, if $\int_{s'} |T(s' \mid \bar{s}, \overline{a}) - T(s' \mid s, \overline{a})| \, ds' \leq \epsilon_T$ and $|R(\bar{s}, \overline{a}) - R(s, \overline{a})| \leq \epsilon_R$, then $|Q^\pi(s, \overline{a}) - Q^\pi(\bar{s}, \overline{a})| \leq \frac{\epsilon_R + \gamma \text{VMAX} \epsilon_T}{1 - \gamma}$. *(Proof in the Appendix.)*

States for which this property does not hold can be refined until they do, as formalized by Algorithm 1, which first constructs an abstract MDP with abstract states for the general state distribution $(\bar{s}_{0,j})$ and every option $i$'s outcome distribution $(\bar{s}_{i,j})$, partitioned by initiation vectors $I_j$. States from which a transition exceeds an error threshold are identified and iteratively refined. The abstract transition and reward functions are computed from data by taking a weighted average of samples after the abstract states are constructed. The abstraction is finite, Markov, and transition-preserving model when no such states remain and is sufficient for planning, after modification for a given *plan query* (see details in Appendix E).

---

**Algorithm 1** Constructing and Refining an Abstract MDP

---

1: **Given:** transitions $\{s_i, I_i, \overline{a}_i, r_i, s'_i, I'_i, t_i\}, \gamma, \epsilon_t, \epsilon_r$
2:
3: ▷ Partition transitions by initiation vector.
4: $\bar{s}_{0,j} \leftarrow \{(s_i, I_i) | I_i = I_j\}, \forall I_j$
5: $\bar{s}_{j,k} \leftarrow \{(s'_i, I'_i) | \overline{a}_i = j, I'_i = I_k\}, \forall j, I_k$
6: $r_{j,k}, t_{j,k} \leftarrow \text{mean}(r_i, t_i | \overline{a}_i = j, I'_i = I_k)$
7:
8: ▷ Construct outgoing edges for executable actions.
9: **for** $\bar{s}, \overline{a}_i$ **where** $\bar{s}.I[i] = true$ **do**
10: $\quad \overline{T}(\bar{s}_{i,n} \mid \bar{s}, \overline{a}_i) \leftarrow (|\bar{s}_{i,n}| / \sum_m |\bar{s}_{i,m}|)$
11: $\quad \overline{R}(\bar{s}, \overline{a}, \bar{s}_{i,n}) \leftarrow r_{i,n}, \overline{\gamma}(\bar{s}, \overline{a}, \bar{s}_{i,n}) \leftarrow \gamma^{t_{i,n}}$
12: **end for**
13: $\overline{S} = \{\bar{s}_{k,l}\} \forall k, l; \overline{A} = \{\overline{a}_i\} \forall i$
14:
15: ▷ Repeatedly refine states with high-error transitions.
16: **while** $\exists \bar{s}, \overline{a}, \bar{s}'$ s.t. model_error$(\bar{s}, \overline{a}, \bar{s}') \geq (\epsilon_t, \epsilon_r)$ **do**
17: $\quad$ refine_state$(\overline{S}, \overline{A}, \overline{T}, \overline{R}, \overline{\gamma}, \bar{s})$
18: **end while**
19: return $\overline{M} = (\overline{S}, \overline{A}, \overline{T}, \overline{R}, \overline{\gamma})$

---

## 3.4 Examples of Learned Abstract MDPs

We demonstrate our approach using a high-dimensional chainwalk domain of length 6 (Figure 3a). The agent is equipped with actions to move left or right to adjacent states, but with 5% probability,

the subsequent state is selected uniformly at random. At each state, the agent observes a sampled $28 \times 28$ MNIST digit [42] (see Appendix A.1 for more detail).

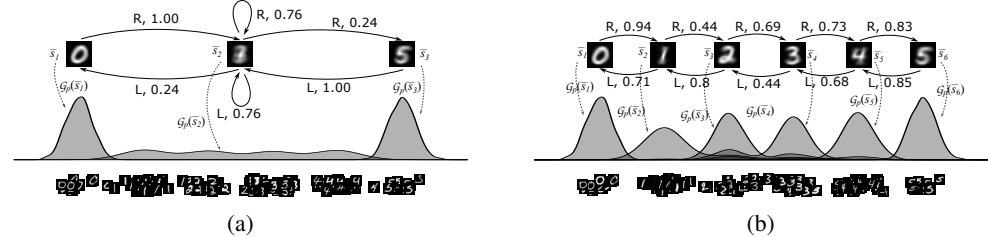

(a)                                        (b)

Figure 3: Learned state abstractions for the MNIST chainwalk. (a) Abstract decision process with three states prior to any refinement. Edges indicate actions with their most likely transitions. Each abstract state is sufficient to determine what actions are executable, and are depicted by their average low-level observations, while grounded samples are shown below. (b) Post-refinement abstract MDP, where the the six underlying discrete states have been recovered.

Algorithm 1 first constructs an abstraction based on the initation vectors (lines 4–13), resulting in a decision process with three abstract states (Figure 3a): the first abstract state grounds to low-level states where only the right action is available, the second to states where both actions are available, and the last to those where only left is executable. Lines 15–18 then iteratively refine these abstract states to recover a 6-state decision process that supports the Markov property (Figure 3b). Model errors are computed using a classifier two-sample test [46], while refinement is achieved by clustering. Implementation details are provided in the appendix.

A simple Gaussian mixture model clustering algorithm is sufficient to refine abstract states in the chainwalk example, but naively clustering more complex observations can be difficult. A common approach in these cases is to first transform the observed data into a more compact form, either by using the features extracted from a pretrained neural network, or explicitly learning a latent space using reconstruction-based objectives. However, such methods do not preserve the Markov property. We therefore use Markov State Abstractions (MSAs) [6], a neural method which approximates provably sufficient conditions to recover Markov representations. A transformation $\phi$ that preserves the Markov property should satisfy the following conditions: the ground and abstract (1) inverse models $I^\pi(a \mid \phi(s'), \phi(s)) = I^\pi(a \mid s', s)$ and (2) their density ratios $p^\pi(\phi(s') \mid \phi(s))/p(\phi(s')) = p^\pi(s' \mid \phi(s))/p^\pi(s')$ are equal. MSAs trained with raw observations output a compressed representation that is approximately Markov, to which Algorithm 1 can be applied.

We apply the resulting neurosymbolic algorithm to a visual Miniworld [17] maze domain. The maze consists of five rooms connected by hallways, and the agent is equipped with options to navigate from the center of rooms to connecting hallways, and vice versa. The north-most room contains a gold block that the agent can approach and pickup. The walls of each room have a unique texture, and the low-level state is given by a $60 \times 80 \times 3$ image of the agent's point of view (Figure 4a, top right). We apply MSA to learn a 8-dimensional representation of the state space, and then apply Algorithm 1. Figure 4a shows the learned abstractions after the refinement, with our approach correctly recovering the underlying abstract graph structure of the environment.

Planning can then take place over this graph using any compatible planner; we ran value iteration on the abstract MDP to compute the goal-conditioned policy.

The planner successfully navigates the agent to west and then north, passing through three hallways, before collecting the gold. We further compare the planning performance of the learned abstract MDP with the goal-conditioned Deep Q-Network (DQN) [48] using the Stable Baselines 3 implementation [51].[3] Figure 4b shows that the abstract MDP achieves better performance with far fewer samples than DQN. In Figure 4c, we increased the observation resolution from $60 \times 80$ pixels to $120 \times 160$ and $180 \times 240$ pixels, while keeping the sample size fixed. As an additional complication, we added random portraits on the wall for the highest resolution—indicated with (P). Although these changes increase both the dimensionality and the complexity of the input space, the underlying decision

---

[3]Most RL libraries assume that every action is available in every state. To make a fair comparison with SB3's DQN implementation, we used the same initiation vector for all ground states during abstract MDP construction.

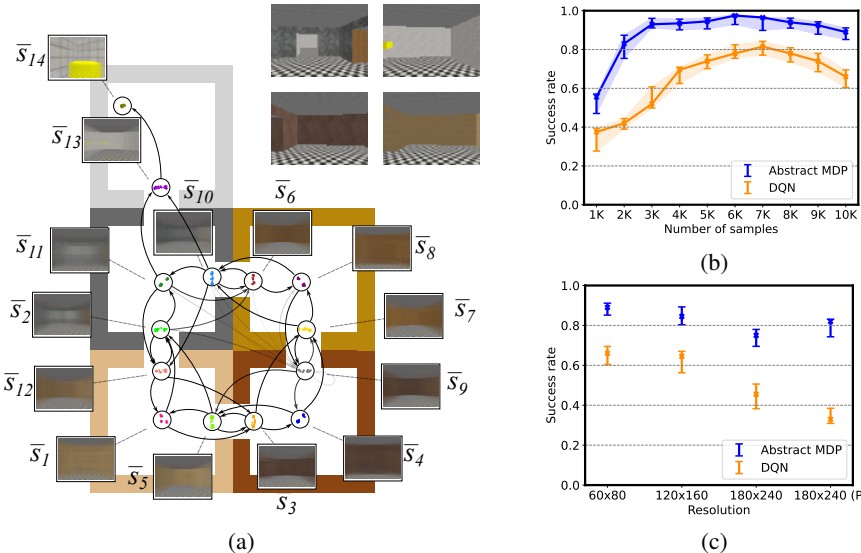

Figure 4: Application of Algorithm 1 with MSA to the visual maze domain. The first abstract MDP is constructed by aggregating states with the same initiation vector. Abstract states are then iteratively refined until there is no improvement on the transition error. (a) Learned abstract MDP consisting of 14 abstract states after the refinement process. Each abstract state is assigned to its own color, while insets show the average ground state at various locations in the domain. The planning performance of methods trained with varying (b) number of samples and (c) resolutions averaged over 20 runs. Shaded regions denote two standard errors around the mean, and the median is marked with a cross.

process remains the same—the correct abstract model should be invariant to the resolution and to the distractor portraits. The results in Figure 4c show that the abstract MDP's performance is not affected by these modifications, whereas the DQN's performance steadily declines with the increasing input complexity. This supports the objective of abstraction: to retain only the information that is necessary for effective decision-making.

## 4 Factored Skills Generate Factored MDPs

The above formulation assumes that option execution always changes every element of the low-level state vector. In practice, agents often have options that change some state variables but not others. Such *factored* options are common in high-dimensional domains like robotics, and we now show that factored options with a subgoal property lead to factored abstract states. Since abstract states ground to distributions over low-level states, we formalize a factored option as one where execution changes the distribution over some state variables, but leaves the distribution over others unchanged (after marginalizing out the modified variables). Formally:

**Definition 2.** An option $o$ is factored with *mask* $m$ from distribution $\bar{s} = P(s[m], s[\overline{m}] \mid I_O)$ if $T(s'|s \sim \bar{s}, o, I'_O) = P(s'[m]|s \sim \bar{s}, o, I'_O) \times P(s[\overline{m}]|I_O)$, where $m$ are the masked state variables, $\overline{m}$ are those *not* in $m$, and $s[m]$ and $s[\overline{m}]$ are the corresponding components of $s$.

Executing factored option $o$ from $\bar{s}$ with mask $m$ differs in its effects on the masked and unchanged state variables. The new distribution on masked variables depends on the start state (drawn from $\bar{s}$) and post-execution initiation vector $I'_O$. The distribution over the unchanged variable is simply $\bar{s}$ with masked variables marginalized out. These two distributions are independent; the variables in $\overline{m}$ are unchanged by $o$ so $s'$ and $I'_O$ do not inform their distribution, while those in $m$ already depend on $s$, so $s[\overline{m}]$ is redundant. This definition can be naturally extended to multiple possible factored outcomes, each with their own mask $m_i$:

**Definition 3.** An *outcome* is a tuple $\omega = (m, I'_O, P(s[m]))$, where $m$ is a mask, $I'_O$ is an observed initiation vector, and $P(s[m])$ is a distribution over the state variables in the mask. Outcome $\omega_i$ results from executing option $o$ from distribution $\bar{s}$ when: only the state variables in $m$ change; the

agent observes subsequent initiation vector $I'_O$; and $P(s[m])$ describes the subsequent distribution over the modified state variables, i.e., $P(s'[m] \mid o, \bar{s}, I'_O)$.

The condition supporting a model-preserving discrete abstract state space also generalizes naturally to factors—that the distribution over modified variables depends on $\bar{s}$ but not low-level state $s \sim \bar{s}$:

**Definition 4.** A factored option $o$ with mask $m$ is a *factored subgoal option* from distribution $\bar{s} = P(s \mid I_O)$ if: $P(s'[m] \mid s \sim \bar{s}, o, I'_O) = P(s'[m] \mid \bar{s}, o, I'_O)$.

## 4.1 Defining a Factored Abstract MDP

The abstract MDP for the factored case must be built out of the masked distributions $P_i(s'[m_i] \mid \bar{s}, o, I'_{Oi})$ that result from factored subgoal option execution. These distributions are only defined over a *subset* $m_i$ of state variables in the ground MDP, and fulfilling any of the Markov properties requires representing a distribution over *all* state variables. We must therefore model how each factored subgoal option modifies a subset of the low-level state variables, while representing a distribution over all of them. We achieve this by grouping the low-level state variables into factors, where an abstract state assigns a factor value to every factor and grounds to a distribution over all low-level state variables, but option execution changes only some of these values.

**Definition 5.** An *atomic factor* $f \subseteq S$ is a group of low-level state variables that are either unmodified by option execution or are all modified together, i.e., $(m_i \cap f \neq \emptyset) \implies f \subseteq m_i$, for all outcome masks $m_i$. Let $F = \{f_1, ..., f_n\}$ be the set of all such factors for a low-level MDP.

An outcome "sets" some of these factors—those inside its mask—to an abstract value representing its masked distribution. We therefore represent each factor as an abstract state variable that can be assigned one of a set of *factor values* representing an outcome $\omega_i$ that includes that factor.

**Definition 6.** A factor $f_j$ has a *factor value* $e_{j,i}$ for each outcome $\omega_i$ for which $f_j \subseteq m_i$, i.e., for each outcome that changes the variables in the factor.

An abstract state is then an assignment of values to every factor, representing a distribution over every low-level state variable and an observed initiation vector.

**Definition 7.** Given a ground MDP with factored subgoal options and $n$ factors, an abstract state $\bar{s}$ is an assignment of exactly one factor value $e_j \in E_j$ to each factor $f_j$: $\bar{s} = [e_1, ..., e_n]$, and an initiation vector $\bar{s}_g = I_O$. The abstract state space $\overline{S}$ is the set of all such assignments.

Transitions in the abstract MDP set the factors overlapping the relevant outcome's mask to the factor values representing its effect; other factor values remain unchanged. The remaining decision process construction—which satisfies all three Markov properties—follows straightforwardly, as does the learning algorithm (both are detailed in the Appendix). Abstract states are grounded as:

**Definition 8.** The state distribution grounding of factored abstract state $\bar{s}$ is the product of the subgoal distributions for each outcome with effects present in $\bar{s}$, after the factors belonging to each outcome absent in $\bar{s}$ have been integrated out. Formally:

$$\mathcal{G}_p(\bar{s}) = \Pi_{\omega_i \in \omega(\bar{s})} \left[ \int \cdots \int_{\bar{e}(\omega_i, \bar{s})} P_i(s[m_i]) d \cdots d_{\bar{e}(\omega_i, \bar{s})} \right],$$

where $\omega(\bar{s}) = \{\omega_i | \exists j \ \bar{s}[j] = e_j(\omega_i)\}$ is the collection of outcomes with an effect value in $\bar{s}$, and $\bar{e}(\omega_i, \bar{s}) = \{e_j(\omega_i) | \bar{s}[j] \neq e_j(\omega_i)\}$ are all of $\omega_i$'s effect values that are *not* present in $\bar{s}$. The initiation set grounding is simply the available initiation vector: $\mathcal{G}_I(\bar{s}) = \bar{s}_g$.

**Theorem 4.1.** *Given abstract factored option $\bar{a}$ and abstract state $\bar{s}$ with grounding $\mathcal{G}_p(\bar{s})$, $T(\bar{s}'|\bar{s}, \bar{a})$ grounds to the distribution over states resulting from executing $\bar{a}$ from a state drawn from $\mathcal{G}_p(\bar{s})$. (Proof in the Appendix.)*

## 4.2 Examples of Learned Factored Abstract MDPs

Figure 5a shows the resulting factored abstract MDP on a gridwalk domain, an extension of chainwalk where there are two integers representing the underlying state in each axis, and actions are the cardinal directions that change the state only in one axis. As a result, the learned abstract states, $\{\bar{s}_1, \ldots, \bar{s}_{36}\}$ are now defined as the assignment of a factor value to each factor, $\bar{s} = [e_{1,i}, e_{2,j}]$, over six different

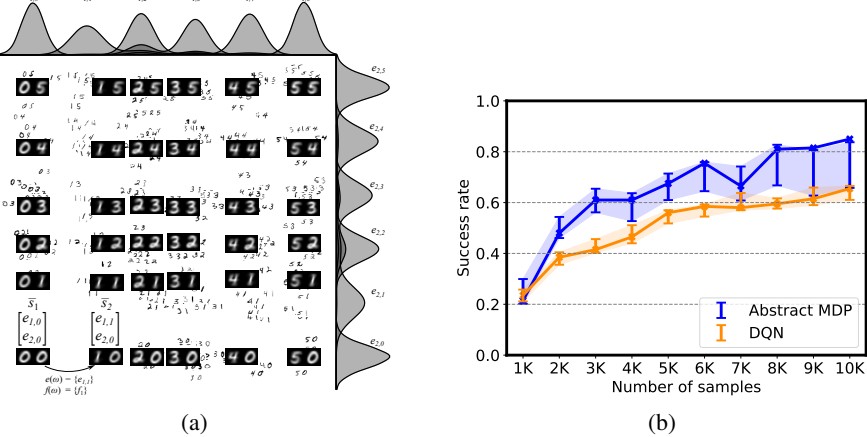

Figure 5: Learned factored abstract MDP on MNIST gridwalk. (a) Abstract states correspond to factored distributions over two factors. Observations are a concatenation of two $28 \times 28$ sized digit images, representing the state in each axis. (b) Planning performance for varying number of samples. Shaded regions denote two standard errors around the mean, and the median is marked with a cross.

values for each factor. Similar to the previous experiment, we compare the planning performance with the goal-conditioned DQN with hindsight experience replay [8] for varying number of samples. We trained an MSA network with 2-dimensional representations on which we can apply Algorithm 1. We added an auxiliary loss to the original MSA objective to minimize the mutual information between hidden units, estimated separately with InfoNCE [67], which allows us to approximate the hidden units as factors. Figure 5b shows that the abstract MDP performs better with fewer samples when compared with DQN. The high variance in abstract MDP performance is due to suboptimal factorization of the MSA embedding space in some runs, which leads to poor refinements. Note that the median values are close to the upper bound of the two standard error bars, suggesting that the factored refinement works well only when the embedding space is properly factored. As such, learning a Markov representation that is factored with respect to the abstract actions would directly improve the task performance [52].

We next apply the algorithm to the challenging Atari game *Montezuma's Revenge* [15]. The state space is given by the annotated RAM states [7] corresponding to 14 factors over 16 variables, while the agent is equipped with a set of factored subgoal options, such as climbing down ladders (see the appendix for the full list), which change only a subset of the RAM variables. We use an expert policy that completes the first room 10% of the time, and otherwise deviates from the plan and executes random actions. This privileged process can be replaced by an exploration module that visits abstract states with fewer samples (or higher transition errors) to expand the frontier of the abstract MDP by further refinement, which we leave as future work. Figure 6 shows visualizations for some of the factored distributions arising from the learned abstraction. After the final refinement, there are 116 factor values over 14 factors that resulted in 1816 abstract states. We then apply value iteration to the resulting abstraction, with the agent learning a policy capable of collecting the key and exiting the room. Figure 6 shows the trajectory of the policy's execution.

## 5 Related Work

Our work builds on constructive approaches in robotics [40, 29] for learning classical planning representations. That work does not result in Markov representations and can only reason about open-loop, straight-line plans. Others have learned object-centric classical planning representations from pixels using deep networks [65, 66, 5, 10] but without any formal guarantees, or using program synthesis [30, 25, 58], which faces scaling challenges. Earlier work in hierarchical RL [20, 34, 22] employed abstract states with hierarchical actions, but they primarily focused on learning policies within fixed hierarchies rather than learning the abstractions themselves.

Aggregation-based state abstractions can be organized into a hierarchy based on whether they preserve specific properties ranging from optimal policies to rewards and transition dynamics for all policies

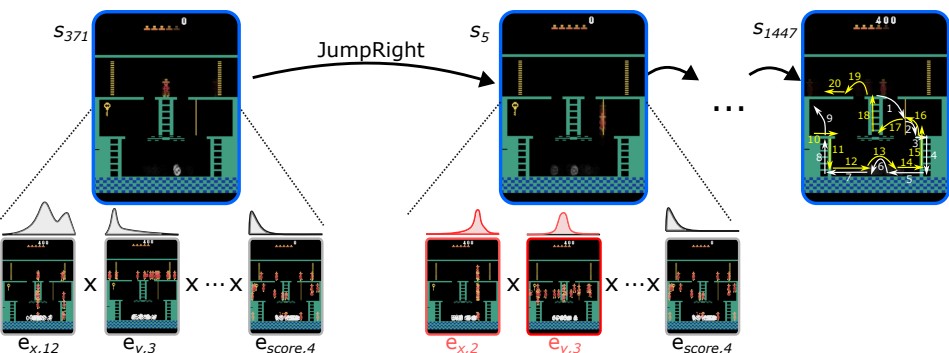

Figure 6: Examples of factored distributions on *Montezuma's Revenge*. Each abstract state (in blue) is an assignment of one factor value (in gray). A factored subgoal option changes some of these factor assignments while leaving the others unmodified. In this example, `JumpRight` action changes factors that correspond to the agent's $x$ and $y$ location and the skull's $x$ location. The execution of the simulated plan that maximizes the reward by applying value iteration is shown on the right.

[44]. The strictest class of abstractions is commonly called model-preservation or bisimulation [21]. Abel [2] provides bounds for $(\phi, \mathcal{O}_\phi)$ state-action abstractions, one of which is similar to Theorem 3.1. Our main difference from these previous works is that we define abstract states as distributions over ground states, as opposed to sets. Even though soft state-aggregation-based methods [59, 9] allow defining distributions, because their abstraction is a function from the ground state to the abstract, supports of different abstract states cannot overlap. Furthermore, these abstractions are not tied to any specific semantics of the ground MDP, and thus, can arbitrarily fail to take into account the shape of the ground distribution. Defining abstract states as distributions over ground states—as the primary structure—supports the recursive computation of the Bellman equation over the abstract policy. State groundings can overlap and have different density functions. This difference in abstract state semantics is a fundamentally different direction in state abstraction.

There is a rich line of work that uses neural networks to learn state abstractions independently from action abstraction, typically by converting intuitively desirable properties into loss functions. These include compression [47], next-state prediction [68], inverse dynamics modeling [18], successor features [13], robotic priors [31], contrastive learning [67], or combinations of these [4, 56, 26]. Allen [6] surveys these techniques and characterizes the specific combination that guarantees the Markov property. The result is MSA, which is compressed but not abstract or skill-driven.

A few works have constructed abstract actions based on abstract states, typically partitioning low-level states using graph-based clustering and constructing options to transition between adjacent clusters [37, 69, 41, 33, 61, 55, 54, 24]. These approaches assume that options can always be constructed between adjacent generated abstract states. That assumption fails in many tasks because options are subject to environment dynamics whereas state abstractions are not. Finally, two recent approaches perform skill-driven state abstraction. Bagaria et al. [12, 11] construct options to form a connected graph and uses the nodes as abstract states. Shah et al. [53] use an option's value function evaluated at a low-level state as an abstract state variable. Neither approach guarantees the Markov property or value preservation.

# 6 Conclusion

Hierarchical RL has always focused on abstract actions to avoid the complexity of generating very long sequences of low-level actions. But fully realizing the promise of hierarchy requires *combining* state and action abstractions, to reformulate a complex agent-level MDP into a entirely new, and potentially much simpler, abstract MDP. Such task-level abstraction is a promising approach to realizing generally-intelligent agents [23, 38], which must necessarily have very complex sensorimotor spaces, and mimics the human ability to form and exploit abstract task representations [28]. Our work combines principled theoretical approaches with the power of neural networks to provide a powerful framework for constructing abstract MDPs.

## Acknowledgments

This research was supported by the ONR under REPRISM MURI N00014-24-1-2603 and grant N00014-22-1-2592, and in part by the NSF Graduate Research Fellowship (Grant No. 2040433). Cameron Allen was supported by a gift from Open Philanthropy to the Center for Human-Compatible AI at UC Berkeley, and an AI2050 Senior Fellowship for Stuart Russell from the Schmidt Fund for Strategic Innovation. Part of this research was conducted using computational resources and services at the Center for Computation and Visualization, Brown University.

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

# A  Environment Details

## A.1  Visual Chainwalk

The visual chainwalk domain consists of six underlying states. Unlike the standard environment, in which the state representation of the agent is a single integer, here the state is a visual image representation of the integer, ranging from 0 to 5. At a given state, the agent receives a $28 \times 28$ greyscale image sampled from the labelled set of MNIST images [42]. At each step, a new image is sampled uniformly at random and flattened into a 784 dimensional vector.

The agent has two options (which are equivalent to its low-level actions, owing to the simplicity of the domain): moving left and right. These options are executable everywhere, except at boundary states where the agent can only move right or left appropriately. With probability $0.95$, option execution is deterministic and the agent transitions to the adjacent state. However, with probability $0.05$, the next state is sampled uniformly at random. For each action, the agent receives a reward of $1$ if it arrives at the goal state, and $0$ otherwise.

## A.2  Visual Maze

The visual maze domain is built in the Miniworld [17] framework and consists of five rooms with hallways connecting the rooms. Figure 7 illustrates the layout of the maze from a top down perspective, along with the $60 \times 80 \times 3$ image of the agent's point of view, which represents the state.

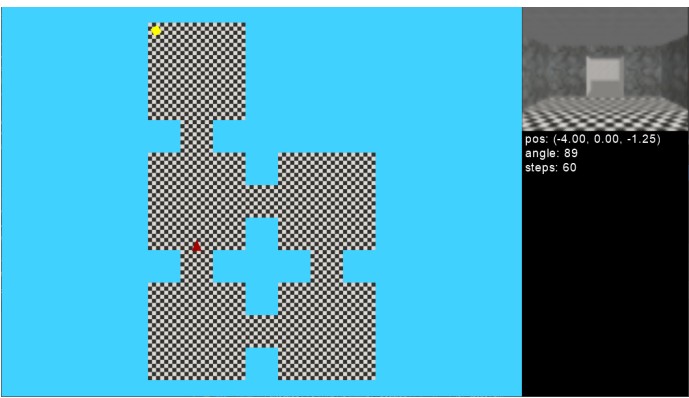

Figure 7: A view of the vault environment, showing the layout of the five rooms and the connecting hallways from a top-down perspective. The agent is represented by a red triangle, and there is a gold block in the north-most room. The state is given by the agent's egocentric view (top right), while additional information such as its exact location is also specified. Note that the state space is only the egocentric pixel observation; no other privileged information is provided to the agent.

Each room has walls with different textures (cardboard, brick, wood, rock and tile) to ensure that the egocentric observation is still Markov. The agent is equipped with five noisy options:

- "Walk to center". This is only executable when the agent is standing in a room's hallway. The option terminates when the agent is in the center of the room.
- "Walk to clockwise hallway". This is executable everywhere, except when the agent is in the north-most room. The option terminates when the agent reaches the hallway closest in the clockwise direction.
- "Walk to anticlockwise hallway". This is executable everywhere, except when the agent is in the north-most room. The option terminates when the agent reaches the hallway closest in the anticlockwise direction.
- "Walk to north hallway". This is only executable when the agent is standing in the center of the room connected to the north-most room. The option terminates when the agent reaches the hallway leading to the room with the gold block.
- "Pickup gold". This is only executable when the agent is standing in the north-most room. The option terminates when the agent has walked to the gold block and picked it up.

The agent receives a constant negative reward for each step taken in the domain, and an episode terminates when the gold block is collected.

### A.3   Visual Gridworld

The visual gridworld is an extension to the chainwalk domain. Here, the state is represented by two MNIST digits, and the agent's actions (options) are each of the cardinal directions (except at boundary states). The two digits represent the underlying $xy$-coordinate on the grid; when the agent transitions to a new state, the image of the unchanged coordinate remains fixed. As such, the state space is of dimension 1568 with are two factors: one for state variables with indices $[0, \ldots, 783]$, and the other for indices $[784, \ldots, 1567]$.

### A.4   *Montezuma's Revenge*

We use the Atari video game *Montezuma's Revenge* as an example of a challenging RL domain. Here, the state space is represented by the underlying 16 bytes that make up the subset of the RAM state that controls the game at a particular time [7]. The domain is otherwise standard, except we equip the agent with the following set of options:

- `RunLeftSkill`: the agent runs left until it reaches the end of a platform or wall.
- `RunRightSkill`: the agent runs right until it reaches the end of a platform or wall.
- `ClimbDownRopeSkill`: the agent climbs down the rope until it reaches the ground.
- `ClimbUpRopeSkill`: the agent climbs up the rope until it reaches the ground.
- `DropFromRopeSkill`: the agent jumps off the rope to the ground.
- `ClimbDownLadderSkill`: the agent climbs down the ladder until it reaches the ground.
- `ClimbUpLadderSkill`: the agent climbs up the ladder until it reaches the ground.
- `JumpInPlaceSkill`: the agent jumps vertically.
- `JumpLeftSkill`: the agent jumps left.
- `JumpRightSkill`: the agent jumps right.
- `WaitForSkullMovingTowards`: the agent stands in place until the skull starts moving towards it.
- `WaitForSkullMovingAway`: the agent stands in place until the skull starts moving away from it.
- `WaitForJumpSkullMovingTowards`: the agent waits in place and the jumps when the skull is moving towards it.
- `WaitForJumpSkullMovingAway`: the agent waits in place and the jumps when the skull is moving away from it.
- `PassToLeftOfJumpSkull`: the agent moves left away from the skull.
- `PassToRightOfJumpSkull`: the agent moves right away from the skull.
- `ChargeEnemyLeft`: the agent runs left towards the skull.
- `ChargeEnemyRight`: the agent runs right towards the skull.

## B   Algorithms for Computing the Model Error and Refinement

### B.1   Using Classifier Two-Sample Test

We compute a surrogate value for $\epsilon_T$ by using a classifier two-sample test (C2ST) [46], which is a popular metric to evaluate generative models that produce distributions over high-dimensional vectors such as natural images. Here, we essentially want to evaluate the discrepancy between $T(s' \mid s, \bar{a})$ and $T(s' \mid \bar{s}, \bar{a})$, which is equivalent to the discrepancy between the joint probability $p(s', s)$ and the product of marginals $p(s')p(s)$ under the expectation of $\bar{s}$. We fit a k-nearest neighbor classifier with $(s, s')$ as positive examples and their shuffled version $(\tilde{s}, s')$ as the negative ones. We measure

the deviation from 50% accuracy, at which $s$ and $s'$ are independent from each other and that the classifier cannot differentiate $(s, s')$ from $(\tilde{s}, s')$. We summarize this procedure in Algorithm 2. During refinement, we sum $\epsilon_T$ values of all abstract states to come up with a cumulative transition error metric $\epsilon_{T_{\text{cum}}}$ for the whole abstract state space. We then compute $\epsilon_{T_{\text{cum}}}$ $m$ times and make a $t$-test to measure the significance of the refinement and accept those that are below a certain value.

---

**Algorithm 2** Compute State Error

---

1: **Given:** $\bar{s}$, transition tuples $\mathcal{D} = \{s, \bar{a}, r, s'\}$ corresponding to $\bar{s}$, factor $f_i$, number of tests $n$, number of nearest neighbors $k$
2:
3: $\epsilon_T, \epsilon_R \leftarrow 0$
4: **for** each available option $\bar{a}_i$ **in** $\bar{s}$ **do**
5: $\quad (s_a, s'_a, r) \leftarrow$ filter_transitions_by_option$(\mathcal{D}, \bar{a}_i)$
6: $\quad m \leftarrow f_i.\text{variables}$
7: $\quad \mathcal{H}_0 \leftarrow \{\}$
8: $\quad \mathcal{H}_a \leftarrow \{\}$
9: $\quad$ **for** $i = 1$ **to** $n$ **do**
10: $\quad\quad x_{\text{ground}} \leftarrow (s_a, s'_a[m])$
11: $\quad\quad x_{\text{abstract}} \leftarrow (s \sim \bar{s}, s'_a[m])$
12: $\quad\quad x_0 \leftarrow (\text{shuffle}(s_a), s'_a[m])$
13: $\quad\quad \mathcal{H}_0 \leftarrow \mathcal{H}_0 \cup \text{knn\_accuracy}(x_{\text{ground}}, x_0, k)$
14: $\quad\quad \mathcal{H}_a \leftarrow \mathcal{H}_a \cup \text{knn\_accuracy}(x_{\text{abstract}}, x_0, k)$
15: $\quad$ **end for**
16: $\quad \mu_0, \sigma_0 \leftarrow$ compute_statistics$(\mathcal{H}_0)$
17: $\quad \mu_a, \sigma_a \leftarrow$ compute_statistics$(\mathcal{H}_a)$
18: $\quad \epsilon_T \leftarrow \epsilon_T + p * (-\log(\text{ttest\_p-value}(\mu_0, \sigma_0, \mu_a, \sigma_a, n)))$
19: $\quad \epsilon_R \leftarrow \epsilon_R + p * \text{var}(r)$
20: **end for**
21: **return** $\epsilon_T, \epsilon_R$

---

## B.2 Refinement

We wish to decompose an abstract state into smaller components, and then recompute the appropriate transition dynamics for each of them (both for incoming and outgoing transitions). Since an abstract state represents a distribution over ground states, there are combinatorially many ways the state could be refined. In practice, we adopt a clustering approach, with each resulting cluster representing a component distribution. We find that expectation-maximization on a Gaussian mixture model with two components is sufficient for the domains presented here, but the refinement process is agnostic to this. Earlier version of this work used $k$-means with $k = 2$ that have similar results. We later adopted GMMs to accommodate overlapping supports. Algorithm 3 illustrates the procedure for refining a given abstract state.

---

**Algorithm 3** refine_state

---

1: **Given:** $\overline{S}, \overline{A}, \overline{T}, \overline{R}, \overline{\gamma}, \bar{s}$, transition tuples $\mathcal{D}$, clustering algorithm $\mathcal{C}$
2:
3: $\{s, \bar{a}, r, s'\} \leftarrow$ filter_transitions_by_abstract_state$(\mathcal{D}, \bar{s})$
4: $\mathcal{C}.\text{fit}(s)$
5: $\text{labels}_s \leftarrow \mathcal{C}.\text{predict}(s)$
6: $\text{labels}_{s'} \leftarrow \mathcal{C}.\text{predict}(s')$
7: Create new abstract states $\overline{S}_{\text{new}}$ based on unique labels from $\text{labels}_s$ and $\text{labels}_{s'}$
8: Assign each transition $(s, \bar{a}, r, s')$ to the corresponding abstract states in $\overline{S}_{\text{new}}$
9: $\overline{S} \leftarrow \overline{S} \cup \overline{S}_{\text{new}} \setminus \{\bar{s}\}$
10: Recompute abstract transitions $\overline{T}$ based on updated transitions

---

### B.3 Using MSA Density Ratio Loss

We used InfoNCE [67] objective to approximate the density ratio objective in MSA. This objective pushes the model to create a contrast between real transition tuples $(s, s')$ and their shuffled version $(s, \tilde{s}')$. Since this objective provides a lower bound to the mutual information $I(s, s')$, it can be directly used as a proxy for detecting states with high mutual information. Such states can be selected as candidates for refinement.

## C   Proof for Theorem 3.1

Let abstract states $\bar{s}$ represent distributions over ground states, defined by $\Pr(s|\bar{s})$. Let $\bar{T}(\bar{s}'|\bar{s}, \bar{a})$ and $\bar{R}(\bar{s}, \bar{a})$ be a transition and reward functions defined over abstract states. Thus, we define $T(s'|\bar{s}, \bar{a}) = \sum_{\bar{s}'} \bar{T}(\bar{s}'|\bar{s}, \bar{a}) \Pr(s'|\bar{s}')$, and similarly define $\pi(\bar{a}|\bar{s}) = \int_s \Pr(s|\bar{s}) \pi(\bar{a}|s)$. Additionally let $R(s, \bar{a}) \geq 0$. VMAX represents the maximum value obtained in either the ground or abstract MDP: $\text{VMAX} = \max\left(\max_s V(s), \max_{\bar{s}} V(\bar{s})\right)$. For conciseness, we define $T^{s,\bar{a},s'} = T(s'|s, \bar{a})$.

**Theorem C.1.** *For all $\bar{s}$, for any $\bar{a}$, for any policy $\pi : S \to \bar{A}$, and any $s \in \{x \in S : P(x \sim \bar{s}) > 0\}$, if*

$$\int_{s'} |T(s' \mid \bar{s}, \bar{a}) - T(s' \mid s, \bar{a})| \, ds' \leq \epsilon_T, \tag{4}$$

$$|\bar{R}(\bar{s}, \bar{a}) - R(s, \bar{a})| \leq \epsilon_R, \tag{5}$$

*then*

$$|Q^\pi(s, \bar{a}) - Q^\pi(\bar{s}, \bar{a})| \leq \frac{\epsilon_R + \gamma \text{VMAX} \epsilon_T}{1 - \gamma}. \tag{6}$$

*Theorem 3.1.* This result is closely related to the *simulation lemma* [45, 35], generalized to apply to abstract state and action spaces. Without loss of generality, we present the proof for the case of $Q^\pi(s, \bar{a}) \geq Q^\pi(\bar{s}, \bar{a})$. The same arguments hold for $Q^\pi(s, \bar{a}) \leq Q^\pi(\bar{s}, \bar{a})$. We begin by proving that a bound on $Q$ differences implies the same bound on $V$ differences, in expectation.

**Lemma C.2.** *Let $\Delta_Q$ be the maximum discrepancy for any $\bar{s}, \bar{a}$, and any $s \in \{x \in S : P(x \sim \bar{s}) > 0\}$:*

$$\Delta_Q = \max_{\bar{s}, o, s \sim \bar{s}} Q^\pi(s, \bar{a}) - Q^\pi(\bar{s}, \bar{a}) \tag{7}$$

*Then*

$$\mathbb{E}_{s \sim \bar{s}} [V^\pi(s)] - V^\pi(\bar{s}) \leq \Delta_Q \tag{8}$$

*Proof of Lemma C.2.*

$$\mathbb{E}_{s \sim \bar{s}} [V^\pi(s)] - V^\pi(\bar{s}) \tag{9}$$

$$= \int_s Pr(s|\bar{s}) \int_{\bar{a}} \Pr(\bar{a}|s) Q^\pi(s, \bar{a}) d\bar{a} ds - \int_{\bar{a}} \Pr(\bar{a}|\bar{s}) Q^\pi(\bar{s}, \bar{a}) d\bar{a} \tag{10}$$

$$= \int_s Pr(s|\bar{s}) \int_{\bar{a}} \Pr(\bar{a}|s) Q^\pi(s, \bar{a}) d\bar{a} ds - \int_{\bar{a}} \int_s \Pr(s|\bar{s}) \Pr(\bar{a}|s) Q^\pi(\bar{s}, \bar{a}) d\bar{a} ds \tag{11}$$

$$= \int_s \int_{\bar{a}} Pr(s|\bar{s}) \Pr(\bar{a}|s) \left(Q^\pi(s, \bar{a}) - Q^\pi(\bar{s}, \bar{a})\right) d\bar{a} ds \tag{12}$$

$$= \mathbb{E}_{s \sim \bar{s}} \left[ \mathbb{E}_{\bar{a} \sim s} \left[ Q^\pi(s, \bar{a}) - Q^\pi(\bar{s}, \bar{a}) \right] \right] \tag{13}$$

$$\leq \mathbb{E}_{s \sim \bar{s}} \left[ \max_{\bar{a}} \left( Q^\pi(s, \bar{a}) - Q^\pi(\bar{s}, \bar{a}) \right) \right] \tag{14}$$

$$\implies \mathbb{E}_{s \sim \bar{s}} [V^\pi(s)] - V^\pi(\bar{s}) \leq \Delta_Q \tag{15}$$

$\blacksquare$

Now, using the Bellman Equation for both $Q^\pi(s, \bar{a})$ and $Q^\pi(\bar{s}, \bar{a})$, we set up a recurrence relation in order to prove our bound.

*Proof of Recursive Relationship.*

$$Q^\pi(s, \bar{a}) - Q^\pi(\bar{s}, \bar{a}) \tag{16}$$

$$= R(s, \bar{a}) + \gamma^\tau \int_{s'} T^{s, \bar{a}, s'} V^\pi(s') ds' - R(\bar{s}, \bar{a}) - \gamma^\tau \sum_{\bar{s}'} T^{\bar{s}, \bar{a}, \bar{s}'} V^\pi(\bar{s}') \tag{17}$$

$$= \underbrace{R(s, \bar{a}) - R(\bar{s}, \bar{a})}_{\leq \epsilon_R} + \gamma^\tau \int_{s'} T^{s, \bar{a}, s'} V^\pi(s') ds' - \gamma^\tau \sum_{\bar{s}'} T^{\bar{s}, \bar{a}, \bar{s}'} V^\pi(\bar{s}') \tag{18}$$

$$\leq \epsilon_R + \gamma^\tau \int_{s'} T^{s, \bar{a}, s'} V^\pi(s') ds' - \gamma^\tau \sum_{\bar{s}'} T^{\bar{s}, \bar{a}, \bar{s}'} \underbrace{V^\pi(\bar{s}')}_{\geq \mathbb{E}_{s' \sim \bar{s}'}[V^\pi(s') - \Delta_Q] \ \ \text{(Lemma C.2)}} \tag{19}$$

$$\leq \epsilon_R + \gamma^\tau \int_{s'} T^{s, \bar{a}, s'} V^\pi(s') ds' - \gamma^\tau \sum_{\bar{s}'} T^{\bar{s}, \bar{a}, \bar{s}'} \int_{s'} \Pr(s'|\bar{s}') \left(V^\pi(s') - \Delta_Q\right) ds' \tag{20}$$

$$= \epsilon_R + \gamma^\tau \int_{s'} T^{s, \bar{a}, s'} V^\pi(s') ds' - \gamma^\tau \int_{s'} \underbrace{\sum_{\bar{s}'} T^{\bar{s}, \bar{a}, \bar{s}'} \Pr(s'|\bar{s}')}_{= T^{\bar{s}, \bar{a}, s'}} \left(V^\pi(s') - \Delta_Q\right) ds' \tag{21}$$

$$= \epsilon_R + \gamma^\tau \int_{s'} T^{s, \bar{a}, s'} V^\pi(s') ds' - \gamma^\tau \int_{s'} T^{\bar{s}, \bar{a}, s'} \left(V^\pi(s') - \Delta_Q\right) ds' \tag{22}$$

$$= \epsilon_R + \gamma^\tau \Delta_Q + \gamma^\tau \int_{s'} \left(T^{s, o, s'} - T^{\bar{s}, \bar{a}, s'}\right) V^\pi(s') ds' \tag{23}$$

$$\leq \epsilon_R + \gamma^\tau \Delta_Q + \gamma^\tau \underbrace{\int_{s'} \left| T^{s, \bar{a}, s'} - T^{\bar{s}, \bar{a}, s'} \right|}_{\leq \epsilon_T} \underbrace{|V^\pi(s')|}_{\leq \text{VMAX}} ds' \tag{24}$$

$$\leq \epsilon_R + \gamma^\tau \Delta_Q + \gamma^\tau \epsilon_T \text{VMAX} \tag{25}$$

$$\leq \epsilon_R + \gamma \Delta_Q + \gamma \epsilon_T \text{VMAX} \tag{26}$$

$$\blacksquare$$

Since this holds for all $\bar{s}, o$, this implies that

$$\Delta_Q \leq \epsilon_R + \gamma \epsilon_T \text{VMAX} + \gamma \Delta_Q \tag{27}$$

$$\implies \Delta_Q \leq \frac{\epsilon_R + \gamma \epsilon_T \text{VMAX}}{1 - \gamma} \tag{28}$$

Thus completes our proof.

## D  Proof for Theorem 4.1

To first define the abstraction transition function, we observe that transitions in the abstract MDP modify the factors contained in the relevant outcome's mask, causing the values of factors in the mask to be changed to the factor values representing its effect. All other factor values remain unchanged.

The corresponding abstract transition function is:

$$T(\bar{s}'|\bar{s}, \bar{a}) = \begin{cases} p_i & \text{if } \bar{s}'[f_j] = e_j(\omega_i), \forall f_j \in f(\omega_i) \text{ and} \\ & \bar{s}'[f_k] = \bar{s}[f_k], \forall f_k \notin f(\omega_i) \text{ and} \\ & \bar{s}'_g = I'_{Oi} \\ \\ 0 & \text{otherwise,} \end{cases} \tag{29}$$

where the outcome $\omega_i$ with mask $m_i$ and initiation vector $I'_{Oi}$ occurs with probability $p_i$ (possibly 1) when executing action $\bar{a}$ from $\bar{s}$.

**Theorem D.1.** *Given an abstract factored option $\bar{a}$ and an abstract state $\bar{s}$ with grounding $\mathcal{G}_p(\bar{s})$, then $T(\bar{s}'|\bar{s},\bar{a})$ computed by Equation 29 accurately represents the distribution over states in which the agent may find itself after executing $o$ from a state drawn from $\mathcal{G}_p(\bar{s})$.*

*Proof of Theorem D.1.* Given that abstract state $\bar{s}$ has grounding:

$$\mathcal{G}_p(\bar{s}) = P(s \mid I_O) = \Pi_{\omega_i \in \omega(\bar{s})} \left[ \int \cdots \int_{\bar{e}(\omega_i,\bar{s})} P_i(s[m_i]) d \cdots d_{\bar{e}(\omega_i,\bar{s})} \right], \tag{30}$$

we seek to prove that the abstract transition function grounds to the distribution over states the agent may find itself after executing action $\bar{a}$ from a state drawn from $\mathcal{G}_p(\bar{s})$. By combining multiple factored outcomes and factored subgoal options in Definitions 3 and 4 with Definition 2, we arrive at:

$$T(s' \mid s \sim \bar{s}, o, I_O') = \sum_{\omega_i} p_i \left[ P_i(s'[m_i] \mid \bar{s}, \bar{a}, I_{Oi}') \times P(s[\overline{m}_i] \mid I_O) \right],$$

where there are $i$ outcomes. The abstract transition function has a similar form, via Equation 29: $T(\bar{s}'|\bar{s},\bar{a}) = \sum_i p_i \bar{s}_i'$. It therefore suffices to show that:

$$\mathcal{G}_p(\bar{s}_i') = P_i(s'[m_i] \mid I_{Oi}') \times P(s[\overline{m}_i] \mid I_O), \tag{31}$$

for each outcome $\omega_i$. We begin by applying the grounding definition (Definition 8) to state $\bar{s}_i'$, which gives:

$$\mathcal{G}_p(\bar{s}_i') = \Pi_{\omega_j \in \omega(\bar{s}_i')} \left[ \int \cdots \int_{\bar{e}(\omega_j,\bar{s}_i')} P_j(s[m_j] \mid I_{Oj}) d \cdots d_{\bar{e}(\omega_j,\bar{s}_i')} \right],$$

where $\omega(\bar{s}_i')$ is the set of outcomes with factor values present in $\bar{s}_i'$, and $\bar{e}(\omega_j,\bar{s}_i')$ is the set of factors that are in $\omega_j$'s mask but have factor values not in $\bar{s}_i'$. One outcome present in $\bar{s}_i'$ is $\omega_i$ itself, which we know by Equation 29 has every relevant factor value set in $\bar{s}_i'$:

$$\bar{s}_i'[f_j] = e_j(\omega_i), \forall f_j \in f(\omega_i),$$

and hence $\bar{e}(\omega_i, \bar{s}_i')$ is empty, so:

$$\mathcal{G}_p(\bar{s}_i') = P_i(s'[m_i] \mid I_{Oi}) \times \Pi_{\omega_j \in \omega(\bar{s}')\setminus\omega_i} \left[ \int \cdots \int_{\bar{e}(\omega_j,\bar{s}')} P_j(s[m_j] \mid I_{Oj}) d \cdots d_{\bar{e}(\omega_j,\bar{s}')} \right]. \tag{32}$$

Comparing equations 31 and 32 we see that the left components of the product are the same; it remains to show that the right portions are equal. Specifically, we must show that:

$$P(s[\overline{m}_i] \mid I_O) = \Pi_{\omega_j \in \omega(\bar{s}')\setminus\omega_i} \left[ \int \cdots \int_{\bar{e}(\omega_j,\bar{s}')} P_j(s[m_j] \mid I_{Oj}) d \cdots d_{\bar{e}(\omega_j,\bar{s}')} \right].$$

Since neither term is a function of any variables in $m_i$, we can freely introduce additional integrals over $m_i$. We must therefore prove:

$$P(s[\overline{m}_i] \mid I_O) = \int \cdots \int_{(m_i)} \Pi_{\omega_j \in \omega(\bar{s}')\setminus\omega_i} \left[ \int \cdots \int_{\bar{e}(\omega_j,\bar{s}')} P_j(s[m_j] \mid I_{Oj}) d \cdots d_{\bar{e}(\omega_j,\bar{s}')} \right] d_{(m_i)}.$$

Next, we note that $\omega(\bar{s}')\setminus\omega_i \subseteq \omega(\bar{s})$, i.e., that every outcome in $\omega(\bar{s}')\setminus\omega_i$ is also in $\omega(\bar{s})$. Additionally, outcomes in $\omega(\bar{s})$ but not in $\omega(\bar{s}') \setminus \omega_i$ can only be functions of $m_i$, and so can be included in the product on the right hand side because they will subsequently be integrated out. We must therefore now prove that:

$$P(s[\overline{m}_i] \mid I_O) = \int \cdots \int_{(m_i)} \Pi_{\omega_j \in \omega(\bar{s})} \left[ \int \cdots \int_{\bar{e}(\omega_j,\bar{s}')} P_j(s[m_j] \mid I_{Oj}) d \cdots d_{\bar{e}(\omega_j,\bar{s}')} \right] d_{(m_i)}.$$

Finally, we can replace $\bar{e}(\omega_j,\bar{s}')$ above with $\bar{e}(\omega_j,\bar{s})$ because these terms can only differ via factors in $m_i$, which are subsequently integrated out. Hence it remains to prove that:

$$P(s[\overline{m}_i] \mid I_O) = \int \cdots \int_{(m_i)} \Pi_{\omega_j \in \omega(\bar{s})} \left[ \int \cdots \int_{\bar{e}(\omega_j,\bar{s})} P_j(s[m_j] \mid I_{Oj}) d \cdots d_{\bar{e}(\omega_j,\bar{s}')} \right] d_{(m_i)}$$

$$= \int \cdots \int_{(m_i)} \mathcal{G}_P(\bar{s}) \, d_{(m_i)},$$

which is its definition. $\blacksquare$

# E    Planning using An Abstract MDP

An abstract MDP $\overline{M}$ is sufficient for planning, after modification for a given *plan query*:

**Definition 9.** A *plan query* on the ground MDP is a tuple $Q_\tau = (B_\tau, G_\tau, R_\tau)$, where $B_\tau(s)$ is a start state distribution, $G_\tau \subseteq S$ is a set of terminal goal states, and $R_\tau : G_\tau \to \mathbb{R}$ is the reward for entering states in $G_\tau$.

$\overline{M}$ must be modified as follows: first, execution should cease when the agent enters a state in $G_\tau$, with terminating reward given by $R_\tau$. Any abstract state $\overline{s}$ that overlaps $G_\tau$ must therefore be refined into a terminating abstract state contained in $G_\tau$, and possibly a state representing $\overline{s}$'s remaining support. This process is depicted in Figure 8. The resulting goal states are grouped into goal set $\overline{G}$.

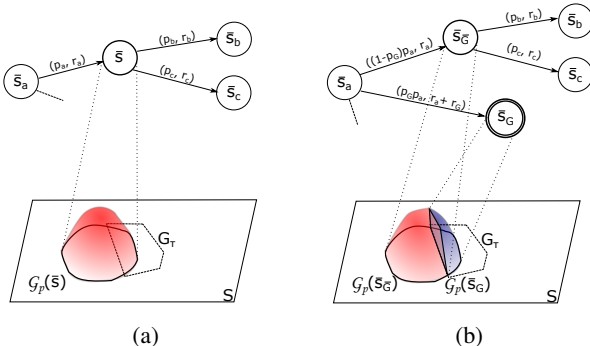

(a)                              (b)

Figure 8: (a) Abstract state $\overline{s}$ with grounding distribution $\mathcal{G}_p(\overline{s})$ that overlaps grounded goal set $G_\tau \subseteq S$. (b) $\overline{s}$ must be refined into a goal state $\overline{s}_G$ that grounds to the portion of $\mathcal{G}_p(\overline{s})$ overlapping $G_\tau$ (with probability $p_G$), and $\overline{s}_{\overline{G}}$, which grounds to the remainder of $\mathcal{G}_p(\overline{s})$ (probability $1 - p_G$). $\overline{s}_G$ is terminating with expected goal reward $r_G$, while $\overline{s}_{\overline{G}}$ has the same reward and outgoing transitions as $\overline{s}$. Incoming transitions to $\overline{s}$ are distributed according to $p_G$.

Second, the agent must collect into start state set $B$ all states with non-empty grounding overlap with $B_\tau$. These need not be refined because the abstraction is approximately value preserving. Any planning algorithm can then be used to compute an optimal policy leading from any state in $B$ to any in $G$.

# F    Limitations

In this work, we derive an algorithm to build an abstract MDP that can be iteratively refined to reduce the transition and reward error to an arbitrary value. Our implementation uses GMMs for such refinements. While partitioning an abstract state essentially decrease its size, and therefore makes it more likely to improve the model, it is possible to hit the local minima with the iterative clustering, and fail to refine further. On the other hand, our method is agnostic to the choice of the clustering approach used, and thus, we believe that this should not pose a serious limitation.

Our experiments in *Montezuma's Revenge* uses annotated RAM variables, While it is possible to train an MSA network on the RGB observations, the learned representations are not necessarily factored, which makes it hard to show how the algorithm operates. While this is not directly a limitation of the method—as our focus is on how to learn factored abstract representations from ground variables from given factors—it can be regarded as the limitation of the paper.

Currently, Algorithm 1 works in a batched fashion on a set of environment interactions. Ideally, it should iteratively collect data from states that are not visited frequently to expand its frontier. As we are interested in constructing the abstract MDP given enough data, we have used an expert policy in 10% of the transitions in *Montezuma's Revenge*, and deviate from the expert policy otherwise.

# G Network Details

```
MSAFlat(
  (encoder): Sequential(
    (0): Linear(in_features=1568, out_features=64, bias=True)
    (1): ReLU()
    (2): Linear(in_features=64, out_features=64, bias=True)
    (3): ReLU()
    (4): Linear(in_features=64, out_features=N_LATENT, bias=True)
  )
  (inverse_fc): Sequential(
    (0): Linear(in_features=4, out_features=64, bias=True)
    (1): ReLU()
    (2): Linear(in_features=64, out_features=64, bias=True)
    (3): ReLU()
    (4): Linear(in_features=64, out_features=5, bias=True)
  )
  (density_fc): Sequential(
    (0): Linear(in_features=4, out_features=64, bias=True)
    (1): ReLU()
    (2): Linear(in_features=64, out_features=64, bias=True)
    (3): ReLU()
    (4): Linear(in_features=64, out_features=1, bias=True)
  )
  (decoder): Sequential(
    (0): Linear(in_features=2, out_features=64, bias=True)
    (1): ReLU()
    (2): Linear(in_features=64, out_features=64, bias=True)
    (3): ReLU()
    (4): Linear(in_features=64, out_features=1568, bias=True)
  )
  (mi): Sequential(
    (0): Linear(in_features=4, out_features=64, bias=True)
    (1): ReLU()
    (2): Linear(in_features=64, out_features=64, bias=True)
    (3): ReLU()
    (4): Linear(in_features=64, out_features=1, bias=True)
  )
)
```

We used a two-layered multi-layer perceptron (MLP) with 64 hidden units and ReLU activations for the MSA encoder. The output dimensionality of the MSA encoder is set to 8 for Visual Maze domain, and to 2 for MNIST gridwalk domain. `inverse_fc` and `density_ec` modules compute the inverse model and the density ratio losses, respectively. `mi` is trained seperately (with a seperate optimizer) to estimate the mutual information between two hidden units, and the main MSA model backpropagates through this estimate to minimize it. Lastly, we use the decoder only for visualization purposes and do not backpropagate any error through it, which would otherwise make it a reconstruction-based embedding.

```
QNetwork(
  (features_extractor): Linear(in_features=1568, out_features=64, bias=True)
  (q_net): Sequential(
    (0): Linear(in_features=4704, out_features=64, bias=True)
    (1): ReLU()
    (2): Linear(in_features=64, out_features=64, bias=True)
    (3): ReLU()
    (4): Linear(in_features=64, out_features=4, bias=True)
  )
)
```

We used the default structure of the DQN model that is provided in stable-baselines3[4] [51] with the only difference in the output dimensionality that depends on the number of actions in the domain. Also, we used the same feature extractor to process the observation and the desired goal, which considerably increased the performance, as opposed to plain concatenation of them. Note that the encoder of MSA and the Q network of the DQN has the same number of layers and hidden units.

## H  Hyperparameters

We used the default hyperparameters of the DQN model in stable-baselines3 with the only change of train frequency from four to one, on which the model was performing better (Table 3). As we are using the goal-conditioned DQN, we experimented with hindsight experience replay [8] and found that while it helped in MNIST gridwalk domain, it made the results worse for the Visual Maze domain. For MSA training, we only did an informal search of frequently used hyperparameter values (Table 1).

<table>
<tr><td colspan="2">Table 1: MSA hyperparameters</td><td colspan="2">Table 2: Alg. 1 hyperparameters</td></tr>
<tr><td>Parameter</td><td>Value</td><td>Parameter</td><td>Value</td></tr>
<tr><td>Batch size</td><td>32</td><td>Minimum transition error $\epsilon_T$</td><td></td></tr>
<tr><td>Validation split</td><td>0.1</td><td>    MNIST chain</td><td>0.1</td></tr>
<tr><td>Optimizer</td><td>Adam</td><td>    MNIST grid</td><td>0.1</td></tr>
<tr><td>    Learning rate</td><td>0.001</td><td>    Visual Maze</td><td>None</td></tr>
<tr><td>    $\beta_1$</td><td>0.9</td><td>    Montezuma's Revenge</td><td>0.1</td></tr>
<tr><td>    $\beta_2$</td><td>0.999</td><td>Minimum $\Delta\epsilon_T$ for refinement</td><td></td></tr>
<tr><td>    $\epsilon$</td><td>$10^{-8}$</td><td>    MNIST chain</td><td>0.5</td></tr>
<tr><td>Negative sampling rate</td><td>10</td><td>    MNIST grid</td><td>0.5</td></tr>
<tr><td>Inverse loss coeff.</td><td>1.0</td><td>    Visual Maze</td><td>0.0</td></tr>
<tr><td>Density ratio loss coeff.</td><td>1.0</td><td>    Montezuma's Revenge</td><td>0.5</td></tr>
<tr><td>Smoothness coeff.</td><td>1.0</td><td>Minimum reward error $\epsilon_R$</td><td>None</td></tr>
<tr><td>Mutual information reg. coeff.</td><td>1.0</td><td>Minimum $\Delta\epsilon_R$ for refinement</td><td>None</td></tr>
<tr><td>Maximum number of epochs</td><td>200</td><td>$n$ cluster trials</td><td>10</td></tr>
<tr><td>Early stopping patience</td><td>5</td><td>Min. samples for refinement</td><td>10</td></tr>
</table>

The main hyperparameters of Algorithm 1 (Table 2) are minimum transition and reward errors, $\epsilon_T$, $\epsilon_R$, and their relative improvements after a refinement, $\Delta\epsilon_T$, $\Delta\epsilon_R$. Using the transition error sufficed in our domains. $\epsilon_T$ and $\epsilon_R$ define thresholds to consider an abstract state as a candidate for refinement. We used $\Delta\epsilon_T$ and $\Delta\epsilon_R$ to check whether the clustering decreased the cumulative transition and reward errors. In that sense, $\epsilon$ values set the granularity level of the abstract MDP whereas $\Delta\epsilon$ values filter out refinements that does not improve the score due to poor clustering.

In this paper, we focus on the theoretical construction of the framework, and provide an implementation of this idea. We note that this is one specific instantiation. Many parts—ranging from refinement to independence tests—can be realized with various methods. As such, hyperparameters both in this approach and in the compared baseline is open to further optimization for better performance.

## I  Compute Resources

The algorithm presented in the paper first trains an MSA network and builds an abstract MDP on top of the trained embeddings. The training time of the MSA network depends on the dataset size and the available chip configuration. The construction of the abstract MDP consists of iterative clustering and independence test steps. We profiled the current implementation to understand the main bottlenecks. Two components that contributed to the overall time complexity the most are (1) `torch.cdist`, that computes the Euclidean distance between each pair of vectors in two tensors to estimate $\epsilon_T$ using $k$-nearest neighbor based independence test, and (2) the assignment of each ground sample to the abstract states after clustering. We believe these processes can be further optimized with sampling

---

[4]`https://stable-baselines3.readthedocs.io/en/master/`

Table 3: DQN hyperparameters

| Parameter | Value |
|---|---|
| Batch size | 32 |
| Learning start step | 100 |
| Optimizer | Adam |
| Learning rate | 0.0001 |
| $\beta_1$ | 0.9 |
| $\beta_2$ | 0.999 |
| $\epsilon$ | $10^{-8}$ |
| Soft update coeff. ($\tau$) | 1.0 |
| Discount factor ($\gamma$) | 0.99 |
| Train frequency (every $k$ step) | 1 |
| Gradient steps (every $k$ rollout) | 1 |
| Replay Buffer | HER [8] |
| Number of sampled goals | 4 |
| Target update steps | 10000 |
| Exploration fraction | 0.1 |
| Exploration initial $\epsilon$ | 1.0 |
| Exploration final $\epsilon$ | 0.05 |
| Maximum gradient norm | 10.0 |

approximations and with special data structures like $k$-d trees to reduce the time spent on finding nearest neighbors.

We used a GPU cluster to train multiple MSA networks in batch and build the abstract MDP locally on a laptop with Apple M1 Pro chip and 16 GB of unified memory. DQN models are trained locally on this same laptop and another with NVIDIA RTX 4070 GPU and Intel Core Ultra 9 Processor 185H. It is difficult to give accurate numbers on the training times as GPUs on the cluster are shared accross multiple users. Trainings on the MNIST domain are completed within minutes whereas trainings on the Visual Maze took around 5 to 10 minutes at $80 \times 60$ resolution, and 30 to 50 minutes on higher resolutions.

## J   MSA Embedding Visualizations

We visualized the learned embedding spaces in different domains for varying samples sizes in Figures 9, 10, 11, 12, and 13.

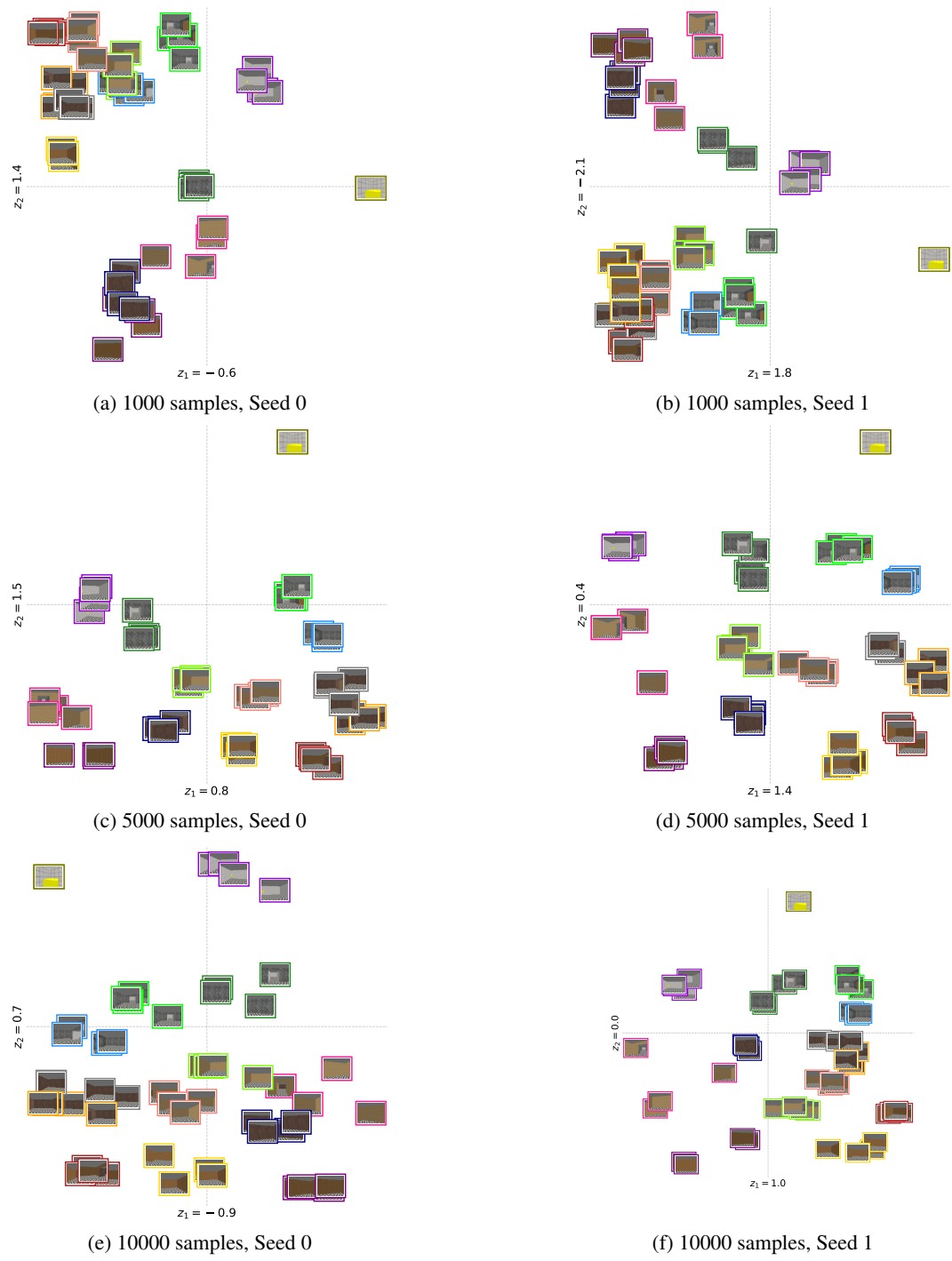

Figure 9: Learned MSA embedding spaces in the Visual Maze domain for different numbers of samples. 2-dimensional positions are the MSA encodings of input images. Frame colors represent the underlying high-level states.

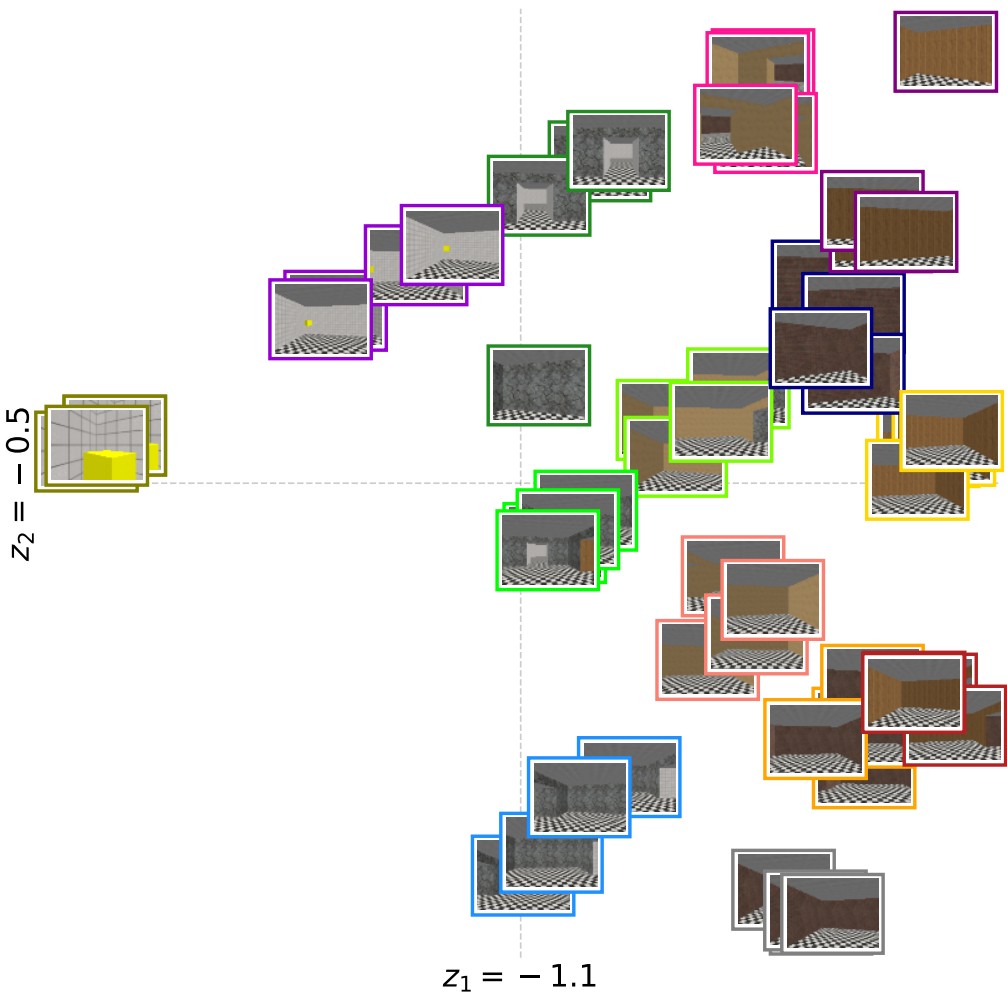

$z_2 = -0.5$

$z_1 = -1.1$

Figure 10: Learned MSA embedding space in the Visual Maze domain at $160 \times 120$ pixels.

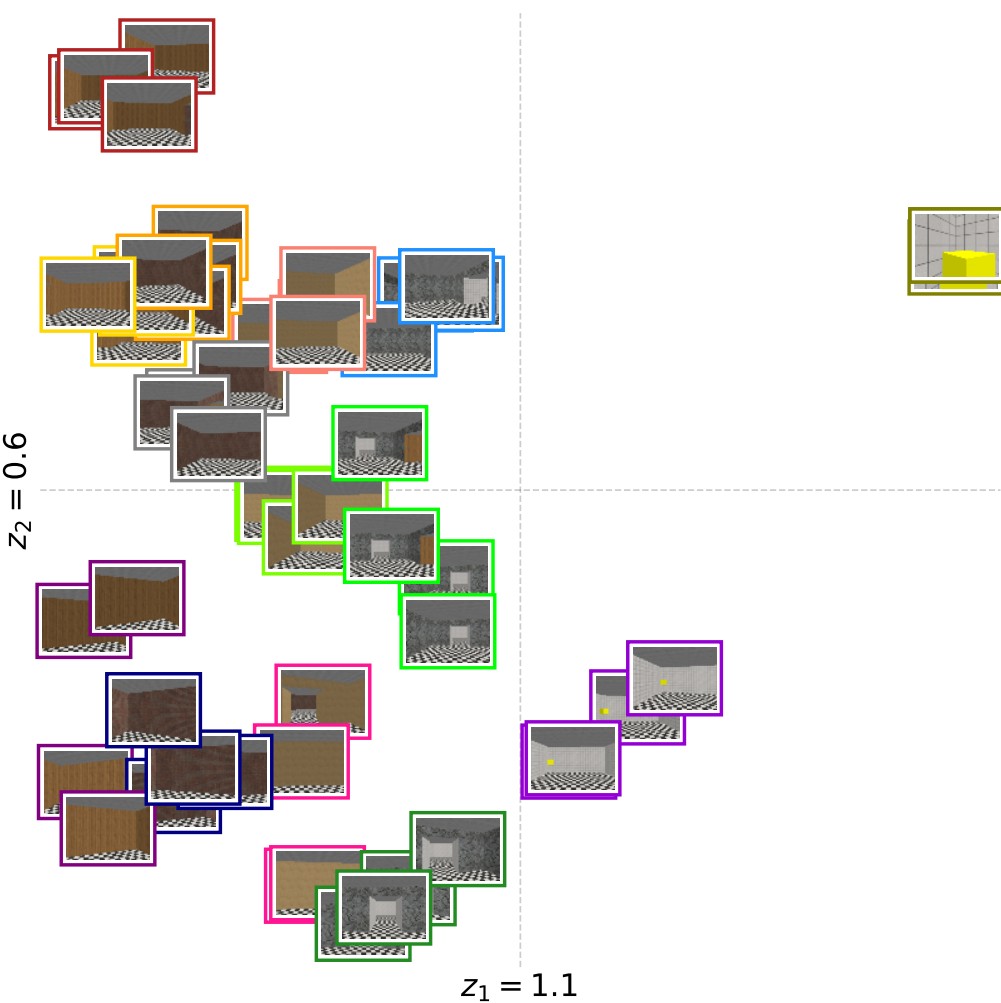

Figure 11: Learned MSA embedding space in the Visual Maze domain at $240 \times 180$ pixels.

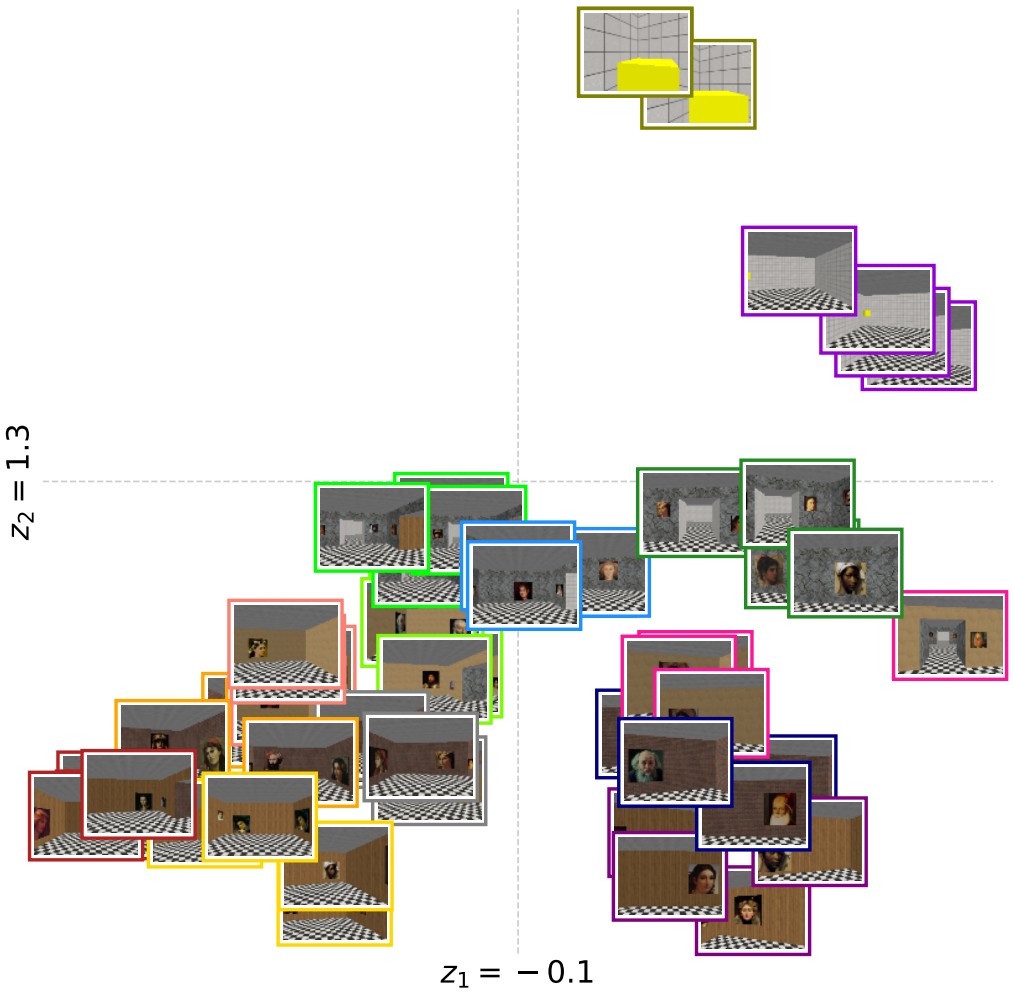

$z_2 = 1.3$

$z_1 = -0.1$

Figure 12: Learned MSA embedding space in the Visual Maze domain at $240 \times 180$ pixels with random portraits on the wall.

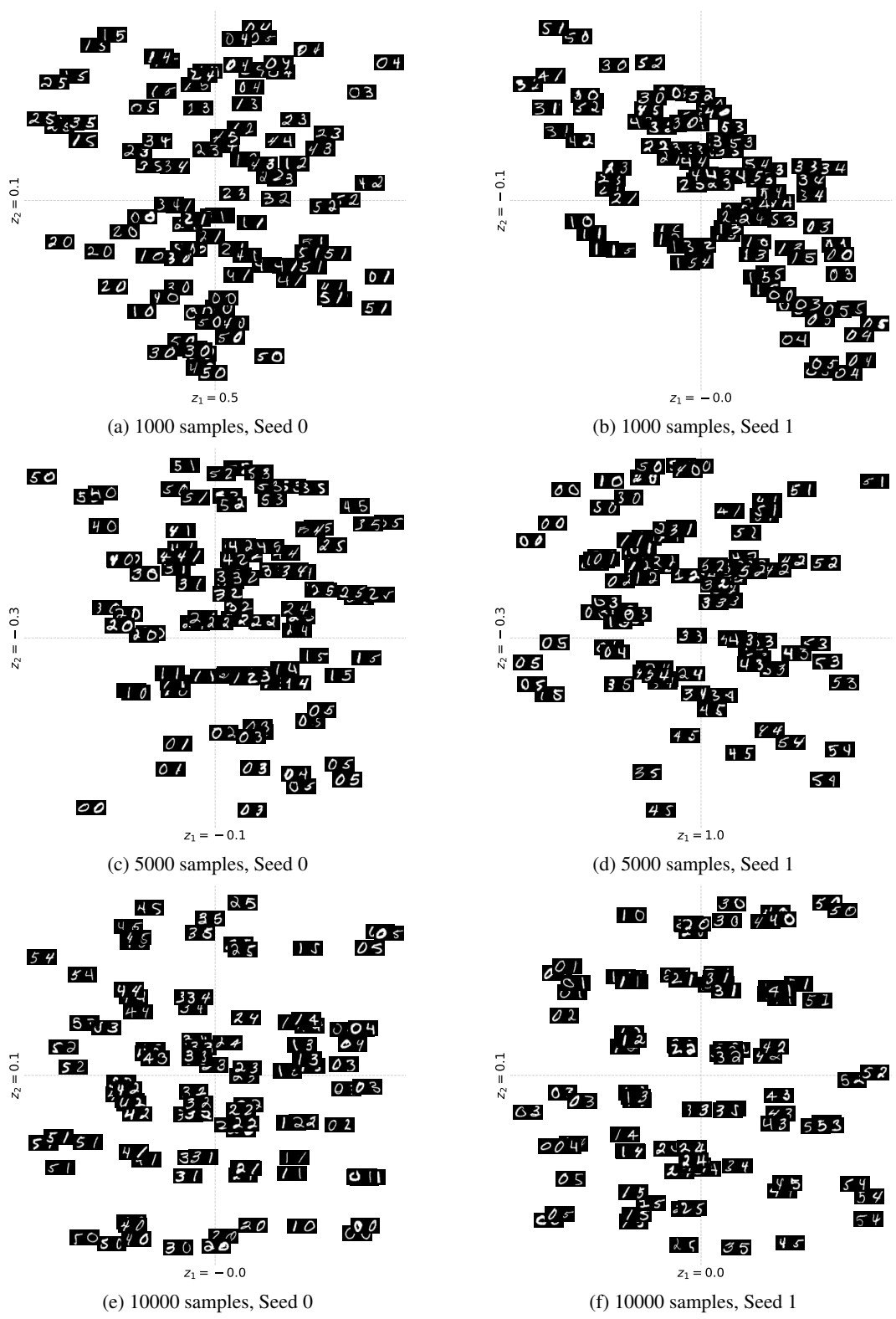

Figure 13: Learned MSA embedding spaces in the MNIST grid domain for different numbers of samples. 2-dimensional positions are the MSA encodings of input images.

