# OpenReview forum: "Skill-Driven Neurosymbolic State Abstractions"
_NeurIPS.cc/2025/Conference — NeurIPS 2025 poster_

### Official Review · Reviewer_buUR · 2025-06-30

**Clarity:** 3
**Significance:** 3
**Originality:** 3
**Rating:** 5
**Confidence:** 3

**Summary:**

This paper attempts to formalize the connection between state abstraction and action abstraction, using probability as a language. More specifically, it discusses the conditions under which, given temporal abstraction (i.e., options), one can find a good state abstraction that aligns with the temporal abstraction.

**Questions:**

Major Concerns:
- The title, introduction, and abstract do not clearly communicate that by “skill-driven,” the authors assume that options (and their associated initiation sets) are given rather than learned. This is a crucial assumption and limitation. It would be helpful to state this more explicitly in the abstract and introduction, so readers can better understand the setting. Similarly, while the current title uses appealing terminology, it does not set the correct expectations for the paper.
- Lines 20–21: I disagree with the claim that current HRL does not address state abstraction. For instance, Leslie Kaelbling’s work on task and motion planning includes a hierarchical structure that clearly resembles state abstraction. Similarly, Peter Dayan’s feudal reinforcement learning explicitly tackles state abstraction. Interestingly, in state-abstraction-oriented work, temporal abstraction is often implicitly assumed through the use of macro actions. However, the reverse is not generally true—i.e., options do not necessarily address state abstraction. It’s valuable that this paper explores ways to bridge that gap.
- It would be enjoyable to see a deeper discussion of the relationship among state abstraction, temporal abstraction, and action abstraction, as the connection is both subtle and important.
- Line 129: I find point 3 confusing. It seems possible to slightly augment the MDP definition to resolve the issue: assume all options are executable everywhere, but selecting an option outside its initiation set returns a huge cost of -100 and consumes no time. Under this formulation, the learned optimal policy should remain the same. Therefore, I do not fully understand why point 3 must hold in order for the process to be Markovian.
- Definition 1: The functions p and I depend on the policy in the previous state. This policy dependency could introduce complications, but it appears under-discussed in the paper. Does this dependency play an essential role?
- Figure 4: I’m confused as to why DQN with hindsight experience replay fails to improve. Its performance seems unchanged from 1K to 10K steps. Why?

Minor Concerns:
- It’s a bit confusing that Equation 1 uses “≈” while Equations 2 and 3 use “=.”
- In Section 3.1, it took me some time to realize that S is being treated as a distribution, which is a shift from how it is handled in Section 2. Although this is mentioned later in the text, it may be helpful to introduce this point earlier—perhaps around line 108—so that readers are better prepared for the formulation that follows.
- Definition 1: I initially interpreted p as a scalar, likely due to the presentation in Figure 2b. But just to confirm—it’s a density, correct?
- Figure 3b: Why do some of the distributions in the middle appear multimodal?

**Ethical Concerns:**

["NO or VERY MINOR ethics concerns only"]

**Final Justification:**

I think it’s a good paper. My score remains unchanged.

**Limitations:**

Yes, but in the appendix. Also, I think some of the theoretical downside (e.g., the connection between state abstraction, temporal abstraction, and action abstraction) is not discussed in detail.

**Paper Formatting Concerns:**

It's fine.

**Quality:**

3

**Strengths And Weaknesses:**

Bridging the option framework to state abstraction is an important goal that has not yet been fully achieved. The approach in this paper is original, and the use of probabilistic state representation is very interesting. It makes a significant and original contribution, and I enjoyed reading it.

Most of my concerns are related to clarity. I’ve outlined them in the Questions section below.

---

> ### Author Rebuttal · Authors · 2025-07-30
>
> Thank you for your comments.
>
> **Title, abstract, and intro do not clearly communicate that skill-driven means given options**
>
> We stated in the abstract and the introduction that we consider constructing state abstractions compatible with a given set of abstract actions, though we might have failed to convey the idea that this approach is fundamentally predicated on the availability of such options, or the importance of this point. We’ll further emphasize this in the introduction.
>
> **Re: Lines 20-21, disagreement with our claim that current HRL does not address state abstraction**
>
> You are right that some earlier hierarchical RL methods (Dayan and Hinton, 1992; Kaelbling 1993; Dietterich 1999) consider a set of state abstractions together with hierarchical actions. While these works also do not focus on learning those state abstractions, our original intention was to refer to works that discover options but do not utilize them for state abstraction (Bacon et al. 2017; Barreto et al. 2019; Machado et al. 2018; Bagaria et al. 2019). Though, re-reading the text, we agree that it is understood as there were no such works, and that we forgot to mention those earlier works as well. We’ll rephrase this statement and explicitly reference them.
>
> **Discussion on the connection between state and action abstraction**
>
> It would definitely be on point to discuss the connection between state and action abstractions in detail. We had to cut off some parts to squeeze in the figures, but we plan to discuss this in the final version. In general, our approach is motivated by the intuition that options are necessarily more constrained than state abstractions—options have a closed-loop interaction with the environment, whereas an agent can hypothesize any state abstraction it wishes—and that combined state-action abstraction should therefore be driven by the properties of its options, with the more flexible state abstraction constructed to match. We will discuss this in more detail in the final version.
>
> **Line 129, point 3, availability of actions everywhere**
>
> In the most general definition, actions might not be available everywhere. From Reinforcement Learning 2nd ed. Sutton and Barto (2018), pg 48:
>
> > Footnote 3. To simplify notation, we sometimes assume the special case in which the action set is the same in all states and write it simply as $\mathcal{A}$.
>
> suggesting $A(s)$ as the general case. That is why we start with $A(s)$ as one of our three target properties. However, the formulation works in the special case where actions are executable everywhere since the refinement process can take care of that (and your suggestion of negative rewards would probably make it easier to work by introducing $\epsilon_r>0$ for those states)—and actually, the experiments in the MNIST grid and the Visual Maze assumed that actions are available everywhere for a fair comparison with the Stable Baselines 3 implementation, which we’ll mention in the updated text. There, the initial abstract MDP consists of a single abstract state that subsumes all states and is refined until a certain threshold is met. Thank you for your suggestion.
>
> **Groundings depend on the policy in Def. 1**
>
> The learned groundings definitely depend on the given options, and in a setting where we learn action abstractions and state abstractions in turns, this would indeed create a complexity. As such, we stand in the position that action abstractions should precede state abstractions.
>
> Regarding the high-level policy (i.e., the exploration policy), we do not think, barring small-sample effects, that it would affect the groundings directly since the termination distribution of the options would stay the same. However, depending on the implementation of Algorithm 1, especially `model_error` and `refine_state`, some abstract states might be of poor quality, or not learned, if they have low visitation counts. Thank you for pointing this out; we’ll clarify it.
>
> **Why DQN with HER fails to improve**
>
> It’s possible that DQN would improve with a better hyperparameter configuration. We didn’t make an extensive hyperparameter search apart from trying some variations with an educated guess. Upon this comment, we re-run the script with more samples, and DQN achieves around 0.5 with 15K samples, and around 0.6 with 20K samples, which is two times longer on the x-axis than our current graph.
>
> **Minor concerns**
>
> Re: Using = in Equations 2 and 3
>
> You are right, Equations 2 and 3 are actually approximations due to the expected-length model (Abel et al., 2019). We’ll fix this. Thank you.
>
> Re: S treated as distribution in Section 3.1
>
> After re-reading the text, we realized that there is some ambiguity to whether we refer to a single probability or a density. For instance, $T(s’ | s, \bar{a})$ can be interpreted as a single scalar referring to the probability of transitioning to a certain $s’$ or a density of $s’$, which might be related to the confusion on Def. 1 as well. Thank you for pointing this out; we’ll make a thorough pass on the used symbols.
>
> Re: p scalar or density in Def. 1
>
> You are correct that $p_i$ in $\mathcal{G}_(\bar{s}_i) = p_i$ is a density whereas $p$s in Figure 2b are scalars, showing how a single distribution can be dissected into a mixture of multiple distributions. We’ll clarify this, thank you.
>
> Re: Figure 3b: multimodal distributions
>
> While the ground distribution is over 28x28 sized images, we visualize them a bit differently. Each sample is assigned with a position equal to its digit plus some noise. Then, to visualize each abstract state, we fit distributions to those position values. While we thought about visualization alternatives that are more faithful to the original semantics of abstract states (e.g., t-SNE), we picked this one as it’s the most pedagogical one. We’ll update the caption accordingly or find some other way to visualize them.
>
> **References**
>
> P. Dayan and G. E. Hinton. Feudal reinforcement learning. *Advances in Neural Information Processing Systems*, 5, 1992.
>
> L. P. Kaelbling. Hierarchical learning in stochastic domains: Preliminary results. In *Proceedings of the Tenth International Conference on Machine Learning*, 951, 1993.
>
> T. Dietterich. State abstraction in MAXQ hierarchical reinforcement learning. *Advances in Neural Information Processing Systems*, 12, 1999.
>
> P. L. Bacon, J. Harb, and D. Precup. The option-critic architecture. In *Proceedings of the AAAI Conference on Artificial Intelligence*, 31, 2017.
>
> M. C. Machado et al. Eigenoption Discovery through the Deep Successor Representation. In *International Conference on Learning Representations*, 2018.
>
> A. Barreto et al. The option keyboard: Combining skills in reinforcement learning. *Advances in Neural Information Processing Systems*, 32, 2019.
>
> A. Bagaria and G. Konidaris. Option discovery using deep skill chaining. In *International Conference on Learning Representations*, 2019.

---

> > ### Comment · Reviewer_buUR · 2025-08-05
> >
> > Thank you for your response.
> >
> > Regarding the policy dependency: I understand your point. It does seem to be a subtle yet important issue, and certain algorithmic designs might help mitigate it. I agree that maintaining a large episodic replay buffer and re-running the clustering algorithm as needed could address the problem, although this approach can be quite computationally expensive. While I remain somewhat unconvinced, I appreciate that the author acknowledges and will discuss the issue—that’s sufficient for me.
> >
> > I have no further questions and look forward to the revised version.

---

> > > ### Author Response · Authors · 2025-08-05
> > >
> > > Thank you for your follow-up. We'll indeed discuss the policy dependency issue and take into account all your other feedback.

---

### Official Review · Reviewer_zE5e · 2025-07-03

**Clarity:** 3
**Significance:** 2
**Originality:** 3
**Rating:** 5
**Confidence:** 2

**Summary:**

The paper presents an approach for learning neurosymbolic abstract Markov decision processes, develops the mathematical foundations and theory of the approach, applies it to both simple and more complex RL problems in synthetic environments, and shows the advantages over vanilla (purely neural) deep reinforcement learning, as in DQN.

**Questions:**

Any further information the authors could give that addresses the weaknesses above would be appreciated!

**Ethical Concerns:**

["NO or VERY MINOR ethics concerns only"]

**Final Justification:**

The rebuttal usefully addresses my comments and I'm happy to raise my score to 5.

**Limitations:**

yes

**Quality:**

3

**Strengths And Weaknesses:**

Strengths: The paper is well motivated and clear, but I am not an expert in RL and so I did not even attempt to follow all the technical details.  It appears technically strong, and a solid implementation of a compelling concept.   The results are reasonable.

Weaknesses:  I am not an expert in the paper's area and so I can't really judge the novelty or level of contribution.  But I'm not sure how it fundamentally advances on earlier work such as https://arxiv.org/abs/2002.05518.   To my naive eye, both the present paper and that earlier work learn a distribution-based representation for abstract states.  The present paper adds an extension to consider learning abstract representations defined by options, but I can't tell how much of a fundamental advance that is.

The paper is strong on theory, but the applications are both quite simple.  The double digit MNIST example is helpful pedagogically but rather contrived; what problems in the real world have such a simple factorization?  The Montezuma's revenge example is more representative of a real world long-horizon planning, but it is still quite simple and the proposed system doesn't solve the whole game from scratch; it starts from a problematic expert policy, and improves it, and it considers only the first room of a long game. The visual maze example is nice, but is still simplified from the real problems of navigation, in ways that play to the algorithm's strength and make it unclear how success there will generalize.

I would also like to understand better how this approach relates to and in practice improves upon other more symbolic ways of learning abstract state and action representations for planning, such as the learned bilevel planners of Silver et al. (AAAI 2022).  That work is briefly mentioned and dismissed here as facing scaling challenges, but it seems that both that work and the present RL-based framework face scaling challenges albeit possibly different ones.

I have to admit that as a non-RL person, I share some of the biases that many people have: RL is nice in theory but just is not a viable approach to building real agents that work in the real world, with reasonable resources.  This paper is elegant but doesn't change my view on this fundamental question, and in some sense only hardens it: we have all this beautiful math, but what does it really buy us?  I recognize this objection may feel unfair to the authors because it targets the whole field of RL, but given the elegance of the work, I feel the authors might be well positioned to answer it.

---

> ### Author Rebuttal · Authors · 2025-07-30
>
> Thank you for your comments.
>
> **Comments regarding the domains**
>
> The point of learning neurosymbolic abstractions is to convert an unnecessarily complex representation of a task into a (much simpler) representation that captures the task’s natural complexity. This is what we do with the MNIST grid and Visual Maze: the intrinsic task dimensionality is only a 6 by 6 grid, but we represent it with two 28 x 28 hand-written digits that change at each step, and likewise for the Maze, the most difficult setting is with 180 x 240 colored images (129,600 features) as observations together with distractor portraits hanging on the wall. In each of these domains, our method recovers the simple natural structure of the task and disregards any additional information. We used domains in which that simple structure is obvious to best convey the idea.
>
> **Difference from Asadi, Abel, and Littman (2020) and other soft state-aggregation-based methods**
>
> They use soft state aggregation (Singh et al., 1995) formulation for the state abstraction and use a neural net to learn these soft weights. We differ from that work in three main points.
>
> First, because their abstraction is a function from the ground state to the abstract, supports of different abstract states cannot overlap. For instance, in a platformer game like Montezuma’s Revenge, jumping to a rope from different sides might result in effect distributions with overlapping supports (in this case, the overlapping part is the agent being on the rope), but the shape of the distribution would be different—it’s more probable that the agent will end up on the rope when it’s jumping from a nearby platform. This is not an edge case that’s specific to the domain but rather a general property that the effects of options would be distributions, and that there is no restriction that would disallow these distributions to overlap.
>
> Second, their learned abstractions are not tied to any specific semantics of the ground MDP, and thus, can arbitrarily fail to take into account the shape of the ground distribution.
>
> Third, those works are primarily theoretical and do not scale to anything like the 10,000-dimensional state domains we have shown here.
>
> Defining abstract states as distributions over ground states immediately allows the recursive computation of the Bellman equation over the abstract policy. State groundings can overlap and have different density functions. This puts our method in a fundamentally new direction in state abstraction.
>
> Both Singh et al. (1994) and Asadi, Abel, and Littman (2020) are quite related, and we will add them to the related work. Thank you for pointing this out.
>
> **Relation to other symbolic approaches; bilevel planning**
>
> The abstract state space in the bilevel planning line of works (Silver et al. 2023) is defined over a fixed grammar, and abstractions are learned through program synthesis, i.e., by a search over the language defined by the grammar. As such, this search requires a set of primitives, which can be pre-defined predicates (Silver et al. 2023) or VLM operations (Liang et al. 2025), which is ultimately a heuristic decision on the structure of abstract states and do not provide any formal guarantees to match the ground MDP. On the other hand, our method does not introduce any such inductive biases on the structure of the abstract states while having theoretical guarantees to match the ground MDP. Finally, our method produces a decision process with the Markov property, which supports the Bellman operator, whereas theirs does not have the Markov property and supports only straight-line planning.
>
> **General RL feedback**
>
> Thank you for sharing your opinion on this. It’s always nice to have such conversations.
>
> We agree that naive reinforcement learning is quite sample-inefficient, which is the motivating reason to learn abstractions. Naively training a DQN agent requires too many interactions, and is simply not a viable approach with reasonable resources, as you also noted.
>
> On the other hand, much of the problem that these methods have arises from the lack of a correct representation. Consider the domains used in the paper. They are simple decision-making problems *once the correct representation is available to the agent*, and this is where our approach kicks in. Now, once you have the action abstractions, you can learn the state abstractions, the form of which aligns with the requirements of value-based RL methods, while using neural nets and still having guarantees on the approximation. This is the first of many papers on this line of work. The only remaining thing would then be learning the action abstractions, which is at least simpler than learning the whole thing from scratch end-to-end. Once this is done, we would have a complete model for planning, effectively converting the RL problem to a model-based planning problem.
>
> Derivations in the paper quantify the degree from which an abstract policy deviates from the ground policy if we were to use these abstractions. It’s reassuring to know that this is bounded by some value, which can be reduced arbitrarily by running the refinement procedure (or with better independence tests and density estimators). In other words, it gives us a way to measure whether the provided abstractions are sufficient. Other derivations verify that we model the correct dynamics in the factored setting. Overall, this type of work gives confidence on the learned representations, helps quantify the error, and makes it easier to figure out the limiting factor.
>
> **References**
>
> S. Singh, T. Jaakkola, and M. Jordan. Reinforcement learning with soft state aggregation. *Advances in Neural Information Processing Systems*, 7, 1994.
>
> K. Asadi, D. Abel, and M. L. Littman. Learning state abstractions for transfer in continuous control. *arXiv:2002.05518*, 2020.
>
> Y. Liang et al. VisualPredicator: Learning Abstract World Models with Neuro-Symbolic Predicates for Robot Planning. In *The Thirteenth International Conference on Learning Representations*, 2025.

---

> > ### Author Response · Authors · 2025-08-08
> >
> > Thank you again for your comments. We hope our response clarified your questions. Please let us know if you have any further questions.

---

### Official Review · Reviewer_JuNa · 2025-07-03

**Clarity:** 3
**Significance:** 3
**Originality:** 2
**Rating:** 5
**Confidence:** 2

**Summary:**

The authors propose an approach to construct state abstractions which are then combined with action-abstraction (options) framework in a hierarchical reinforcement learning (HRL) algorithm. The authors provide some empirical evidence of the effectiveness of their approach.

**Questions:**

What it would take to scale this method up to learning from scratch on Atari?

**Ethical Concerns:**

["NO or VERY MINOR ethics concerns only"]

**Final Justification:**

Strong paper. I think the result-wise, the method could benefit from scaling it up, but it already provides a very solid piece of work. Scaling it up could be viewed as a follow-up.

**Limitations:**

The method does not seem to be scalable yet.

**Quality:**

3

**Strengths And Weaknesses:**

Strengths:
* A novel approach for state abstraction which is combined with action abstraction approach in a theoretically justified way
* Some experimental evidence demonstrating the efficiency of the proposed approach

Weaknesses:
* Regarding experimental efficiency, it is not fully clear whether this approach actually learns well from scratch on Atari Montezuma Revenge, the authors did not provide any learning curves and comparisons with DQNs. I think generally making options-based agents learning efficiently from scratch is a challenging task.
* At the moment, state abstraction seems to be using k-means to come up with these, which is quite similar to [1]. I think this hurts the novelty of the proposed approach.

Minor suggestions:
* I find the notation \bar{s}' quite confusing, in the text between eq.1 and eq.2. From what I read, `\bar{s}'_i` is a probability of s' under mixture component i. Maybe you can change  `\bar{s}'_i` to actually reflect that it is a probability?

References:
[1] Reinforcement Learning with Soft State Aggregation, Satinder P. Singh, Tommi Jaakkola, Michael I. Jordan, NeurIPS 1994

---

> ### Author Rebuttal · Authors · 2025-07-30
>
> Thank you for your comments.
>
> **Scaling up to learn from scratch on Atari**
>
> In this work, we focused on building a principled state abstraction method and showed in the experiments that it can be scaled up to a 180x240x3 dimensional image-based observations. In *Montezuma’s Revenge*, we are given a set of factored options, and show that the algorithm can build the required abstractions to get the agent out of the first room. You are correct that options-based methods require those options to be given, and that learning them from scratch is challenging. Our method would work better if it was combined with a carefully designed exploration algorithm for learning hierarchical abstractions, which is necessary for any sample-efficient RL algorithm, but we leave that to future work. We believe that such a method would reach the end of Montezuma’s Revenge given the options.
>
> **Similarity to soft state-aggregation-based methods**
>
> We differ from soft state-aggregation-based works in three main points.
>
> First, because their abstraction is a function from the ground state to the abstract, supports of different abstract states cannot overlap. For instance, in a platformer game like Montezuma’s Revenge, jumping to a rope from different sides might result in effect distributions with overlapping supports (in this case, the overlapping part is the agent being on the rope), but the shape of the distribution would be different—it’s more probable that the agent will end up on the rope when it’s jumping from a nearby platform. This is not an edge case that’s specific to the domain but rather a general property that the effects of options would be distributions, and that there is no restriction that would disallow these distributions to overlap.
>
> Second, their learned abstractions are not tied to any specific semantics of the ground MDP, and thus, can arbitrarily fail to take into account the shape of the ground distribution.
>
> Third, those works are primarily theoretical and do not scale to anything like the 10,000-dimensional state domains we have shown here.
>
> Defining abstract states as distributions over ground states immediately allows the recursive computation of the Bellman equation over the abstract policy. State groundings can overlap and have different density functions. This puts our method in a fundamentally new direction in state abstraction.
>
> But we agree that Singh et al. (1994) should definitely have been included in Related Work, and will add a citation. Thank you for noticing this.
>
> **Confusion about $\bar{s}’_i$**
>
> We realized that there is indeed some confusion as to whether to interpret $\bar{s}_i’$ as a distribution or a scalar probability, also noted by Reviewer buUR in a different form. We will make this point clear. Thank you for your suggestion.
>
> **References**
>
> S. Singh, T. Jaakkola, and M. Jordan. Reinforcement learning with soft state aggregation. *Advances in Neural Information Processing Systems*, 7, 1994.

---

> > ### Author Response · Authors · 2025-08-08
> >
> > Thank you again for your comments. We hope our response clarified your questions. Please let us know if you have any further questions.

---

> ### Comment · Reviewer_JuNa · 2025-08-08
>
> Thank you for your comments, this clarifies my confusion. I am increasing my score.

---

### Official Review · Reviewer_KwY7 · 2025-07-15

**Clarity:** 3
**Significance:** 3
**Originality:** 3
**Rating:** 5
**Confidence:** 2

**Summary:**

The paper proposes an approach for building a fully abstract MDP from a ground MDP while previous works either focus on action abstraction or on state abstraction only. To cope with this difficult task the authors propose to focus on a strategy that constructs a state abstraction from an action abstraction, for which a number of approaches have been proposed. They generalize their results to the case of factored actions and factored abstract states. Building such MDPs may be viewed as a step towards powerful reasoning machines.
The work is validated on a number of tasks that seem to be built for this evaluation so that comparative results have likely been gained with state of the art strategies run (for instance DQN) on the same tasks. Experimental results show significant improvement of the proposed method over the baselines it is compared to.

**Questions:**

As far as I understand, the peer focuses on the case where one gets a ground MDP and aims at constructing an abstract MDP. Is there any method to directly cope with the construction of an abstract MDP?

Shouldn’t the experimental section compare the proposed method with an alternative method (DQN is not such an alternative)?

A number of elements in the method look ad-hoc. How far one needs to instantiate the method to a particular case to get a relevant result?

**Ethical Concerns:**

["NO or VERY MINOR ethics concerns only"]

**Final Justification:**

I have read the reviewers' reviews and the authors' responses to all the reviews. I'm satisfied with the responses to my comments and those of the other reviewers, which seem convincing to me.
Reading all the reviews/replies, it seems to me that one of the main problems lies in the writing of the paper, with some messages not necessarily sufficiently explicit, and some ambiguities about the notations and the nature of certain variables, which complicate the understanding of the paper. Taking into account all the reviewers' comments according to the authors' answers should significantly improve the paper and this convinced me to increas my score.

**Limitations:**

yes

**Quality:**

2

**Strengths And Weaknesses:**

The paper deals with a difficult task and does a great job at arguing the choices the authors made to achieve their work and at explaining the method in its details. The paper is very well written and structured. Still some parts remain unclear to me but it is likely because I am not an MDP expert. As far as I understand the peer focuses on the case where one gets a ground MDP and aim at constructing an abstract MDP. Is there any method to directly cope with the construction of an abstract MDP?

The experimental section does show clear superiority of the method over baseline on a few tasks on evaluation measure but it does not compare to alternate methods for building an abstract MDP. Shouldn’t the experimental section compare the proposed method with an alternative method (DQN is not such an alternative)?

Part of the contributions are actually described in the Appendix. This is the case for instance of Algorithm 1 which nevertheless looks as a main component of the method. Moreover a number of elements in the method look ad-hoc and need to be chosen by hand by the designer.
For instance the use of Kmeans (section 3.4) and the use of an auxiliary loss in section 4.2. How far is this true, how far one needs to instantiate the method to a particular case to get a relevant result?

In appendix C.1 when you detail the use of a classifier two-sample test, is the k-means what you call the classifier? Couldn’t you use a true classifier, i mean a model learned to discriminate?

I am not sure to understand what is given as input to the method in the examples of section 3.4. As far as i understand from the inputs of the algorithm in Appendix (Algorithme 1), all transitions s_i, I_i, \bar{a}_i, r_i etc are given, meaning the ground MDP?

In Appendix A.3 are all options defined by hand ?

Can you define what “approximately model-preserving” mean? It is mentioned in the paper (for instance page 1 in the introduction).

---

> ### Author Rebuttal · Authors · 2025-07-30
>
> Thank you for your comments.
>
> **Directly constructing an abstract MDP**
>
> We understand your question to be about whether it’s possible to construct an abstract MDP without using the ground MDP, specifically, the $(S, A, T, R)$ tuple. It’s worth noting that our method assumes having access to samples from the ground MDP, not the ground MDP itself. As such, the agent does not have access to the explicit forms of the transition and the reward function—they are rather computed from data by taking a weighted average after the abstract states are constructed. We’ll make this point explicit in the paper, thank you.
>
> **Comparison with an alternative method**
>
> To our knowledge, there is no prior work on constructing an abstract state space that is provably compatible with a given set of action abstractions. As such, there are no similar methods to compare against. The comparison with the de facto value-based RL method, DQN, gives an idea of the overall feasibility and the applicability of the method.
>
> **Some elements in the method look ad-hoc**
>
> We provided a principled method for building abstract states that are *approximately model-preserving* (see below for the newly added definition). Algorithm 1 sketches this construction with two high-level functions `model_error` and `refine_state`. Our method specifies what these functions should do, but is agnostic to how that is accomplished; any suitable independence test for `model_error` or clustering for `refine_state` will do. Any improvement on the components will improve the overall performance of the algorithm. Though you are right; it would be better to include Algorithm 1 in the main text and we’ll try to squeeze it in in the updated version.
>
> **The use of classifier two-sample test**
>
> We used the independence test as a proxy for estimating the model error. Any other method can be used as an independence test, including using a learned discriminator as you suggested. We picked k-NN as it happens to be sample efficient and robust, which was also performing better than trained neural nets for estimating the quality of GAN-generated samples in Lopez-Paz and Oquab (2017). Indeed, as mentioned in Appendix C.3, one can use the density loss of the trained MSA network to detect candidate states for refinement, instead of the classifier two-sample test.
>
> **Input to the method**
>
> Algorithm 1 is for computing $\bar{s}$ from samples collected by executing options $\bar{a} \in \bar{A}$ on the ground MDP. Therefore, the input to the algorithm is these sample transitions, $(s, \bar{a}, r, s’, \tau, I, I’)$, where $\tau$ is the number of steps taken in executing option $o$, and $I$ and $I’$ are the available actions at states $s$ and $s’$, respectively. Moving Algorithm 1 to the main text per your recommendation will make this point clearer. Thank you.
>
> **Options defined by hand**
>
> In the paper, options are constructed by hand, but the method would also apply as-is to learned options. We have deliberately chosen to be agnostic as to how the options are obtained, to make our method compatible with any option discovery method—from hand-coded options to skills learned from demonstration to options obtained from unsupervised RL. After reviewer comments, we realized that it would be better to give a further emphasis on this earlier in the text.
>
> **Approximately model-preserving**
>
> In a model-preserving abstraction, if two ground states $s_1$ and $s_2$ map to the same abstract value, $\phi(s_1)=\phi(s_2)$, then their transition distribution $p(s’ \mid \phi(s_1), a)=p(s’ \mid \phi(s_2), a)$ and rewards $R(s_1, a, s’)=R(s_2, a, s’)$ are equal. In our setting, these are approximately equal up to the errors $\epsilon_t$ and $\epsilon_r$. Note that this definition is not original to our work, it is from Li et al. (2006) and Abel et al. (2016). We’ll define this in the updated text. Thank you.
>
> **References**
>
> D. Lopez-Paz and M. Oquab. Revisiting classifier two-sample tests. In *Proceedings of the Fifth International Conference on Learning Representations*, 2017.
>
> L. Li, T.J. Walsh, and M.L. Littman. Towards a unified theory of state abstraction for MDPs. In
> *Proceedings of the Ninth International Symposium on Artificial Intelligence and Mathematics*,
> 2006.
>
> D. Abel, D. Hershkowitz, and M. Littman. Near optimal behavior via approximate state abstraction. In *International Conference on Machine Learning*, 2016.

---

> > ### Comment · Reviewer_KwY7 · 2025-08-04
> >
> > I have read the reviewers' reviews and the authors' responses to all the reviews. I'm satisfied with the responses to my comments and to those of the other reviewers, which seem convincing to me.
> > Reading all the reviews/replies, it seems to me that one of the main problems lies in the writing of the paper, with some messages not necessarily sufficiently explicit, and some ambiguities about the notations and the nature of certain variables, which complicate the understanding of the paper. Taking into account all the reviewers' comments according to the authors' answers should significantly improve the paper and this convinced me to increase my score.

---

> > > ### Author Response · Authors · 2025-08-04
> > >
> > > Thank you for your thorough validation and reconsideration based on all feedback. We will take into account all the reviewer comments in the final version.

---

### Note · Authors · 2025-08-12

We thank all the reviewers for their careful and thoughtful feedback, and for the follow-up discussions. We believe we have addressed their concerns, and we will incorporate their suggestions to further improve the clarity.

---

### Decision · Program_Chairs · 2025-09-17

**Decision:**

Accept (poster)

**Comment:**

The reviewers all agreed that this work is interesting, technically novel, well-written and that it addresses an important problem (constructing abstract MDPs). It seems to fill a gap in the existing literature, which is to learn state abstractions while being given abstract actions. The empirical results, while not impactful or interesting per se, serve to at least somewhat validate the central claims.

The reviewers have done a good job of highlighting several weaknesses in the paper, many of which the authors have addressed in the rebuttal/discussion.

Overall, the combination of a strong algorithmic contribution combined with a well-executed scholarship lead me to recommend acceptance.